# An invariability-area relationship sheds new light on the spatial scaling of ecological stability

Shaopeng Wang[1,2,3], Michel Loreau[1], Jean-Francois Arnoldi[1], Jingyun Fang[4], K. Abd. Rahman[5], Shengli Tao[4] & Claire de Mazancourt[1]

The spatial scaling of stability is key to understanding ecological sustainability across scales and the sensitivity of ecosystems to habitat destruction. Here we propose the invariability–area relationship (IAR) as a novel approach to investigate the spatial scaling of stability. The shape and slope of IAR are largely determined by patterns of spatial synchrony across scales. When synchrony decays exponentially with distance, IARs exhibit three phases, characterized by steeper increases in invariability at both small and large scales. Such triphasic IARs are observed for primary productivity from plot to continental scales. When synchrony decays as a power law with distance, IARs are quasilinear on a log–log scale. Such quasilinear IARs are observed for North American bird biomass at both species and community levels. The IAR provides a quantitative tool to predict the effects of habitat loss on population and ecosystem stability and to detect regime shifts in spatial ecological systems, which are goals of relevance to conservation and policy.

[1] Centre for Biodiversity Theory and Modelling, Theoretical and Experimental Ecology Station, CNRS and Paul Sabatier University, 09200 Moulis, France. [2] German Centre for Integrative Biodiversity Research (iDiv) Halle-Jena-Leipzig, 04103 Leipzig, Germany. [3] Institute of Ecology, Friedrich Schiller University Jena, 07743 Jena, Germany. [4] Department of Ecology, College of Urban and Environmental Science, and Key Laboratory for Earth Surface Processes of the Ministry of Education, Peking University, 100871 Beijing, China. [5] Forest Research Institute Malaysia, Selangor Darul Ehsan, 52109 Kepong, Malaysia. Correspondence and requests for materials should be addressed to S.W. (email: shaopeng.wang@idiv.de).

Stability is a central problem in ecology. Over the past half century, major advances have been achieved in understanding the stability of local ecological systems and its relationship with biodiversity[1–4]. But sustaining the structure, functioning and services of Earth's ecosystems in the face of global environmental changes requires an improved understanding of stability at large spatial scales. Ecosystem stability at large scales is of particular relevance to human societies as global food security depends on the stability of crop and fish production at regional and global scales[5–8]. The stability of population dynamics at large scales is also critical for the long-term persistence of species[9,10].

Spatial scaling of stability is key to understanding ecological sustainability across scales and predicting the consequences of habitat destruction for long-term ecological dynamics. The idea that stability changes with scale was first acknowledged by Peterson et al.[11], who suggested that the resilience of ecosystems might increase with spatial scale due to scale-mediated effects of diversity. Another study used hierarchical theory to predict that stability (as measured by low variability) at one hierarchical level should increase proportionally to the number of lower-level components[12]. The hierarchical levels used in their study can be interpreted either as discrete biological organizational levels (for example, cell, organ, species) or as spatial scales (for example, local ecosystem, landscape, region). Lastly, using a hierarchical partition of variability across spatial scales, Wang and Loreau[13] predicted that temporal variability should generally decrease with area. However, a quantitative framework has yet to be developed to study the spatial scaling of stability.

Here we propose the invariability–area relationship (IAR) as a novel approach to investigate the spatial scaling of ecological stability. We conceptually define stability as the invariability (the inverse of temporal variability) of a measured ecological variable over time. IAR describes the dependence of population or ecosystem invariability on the area considered. Quantitatively, we measure invariability as the reciprocal of the squared coefficient of variation ($CV^2$) of a population or ecosystem property (for example, abundance, biomass or productivity)[13,14]. This stability metric measures the temporal constancy in the functioning of ecosystems or in the size of populations, and hence in the ecosystem services they might deliver to human societies[15,16]. The spatial scaling of invariability thus informs us about how the reliability of ecosystem services may change across scales.

While mathematicians often define stability as the ability of a dynamical system to return to some state after a small perturbation, ecologists have used a wide array of different stability measures[2,4,17]. Invariability has a number of merits that make it an appropriate starting point to investigate the spatial scaling of stability. First of all, the goal of a stability measure is to quantify the ability of a system to withstand perturbations[4,17]. Stability thus reflects the interplay between intrinsic dynamical processes and perturbations that act upon a system. Therefore, understanding the spatial scaling of stability implies understanding not only the scaling of intrinsic dynamical processes (for example, species interactions, dispersal, and so on) but also that of external perturbations (for example, climate events, fires, and so on). For this purpose, invariability, which measures the magnitude of an ecosystem's response to persistent and erratic environmental perturbations[18] and which can be defined consistently across levels of organization and scales[13,14,19], offers an accessible starting point. Invariability is also easy to quantify in the field, which makes it by far the most commonly used measure of stability in empirical studies[17]. Lastly, recent theoretical studies have demonstrated that invariability is intrinsically related to other measures of stability such as

asymptotic resilience[18,20], Holling's resilience[21], structural stability[22] and persistence[10]. Thus, our study based on invariability may also offer insights into the spatial scaling of other stability metrics.

We start with a simple spatial model to derive IARs in two-dimensional landscapes. This model predicts that IARs are largely determined by patterns of spatial synchrony across scales. In particular, IARs exhibit triphasic curves when spatial synchrony decays exponentially with distance, but are quasilinear when spatial synchrony decays as a power law. We then investigate IARs empirically using long-term continental-scale data of primary productivity and bird biomass. These two data sets, which exhibit triphasic and quasilinear IARs respectively, provide contrasting examples of how different spatial synchrony patterns can generate different IARs. Our work provides a quantitative theory of the spatial scaling of ecological stability. As such it is likely to have significant implications in both ecological research and conservation management. We hope that IAR will open new research prospects similar to the classical species–area relationship (SAR)[23], with potentially as wide and important applications.

## Results

**IARs in model spatial ecosystems.** We use a simple spatial model to derive IARs in two-dimensional landscapes. The landscape consists of regularly distributed square patches, each with unit patch area, in which local biomass (or any other population or ecosystem property) fluctuates due to various ecological factors. For simplicity, we assume that all patches have identical temporal mean ($\mu$) and variance ($\sigma^2$) of biomass, and hence identical variability: $CV_1^2 = \sigma^2/\mu^2$. Between patches, biomass fluctuations can exhibit spatial synchrony due to environmental correlation and/or dispersal[23,24]. We denote $\rho_{x,y}$ as the temporal correlation between patches $x$ and $y$, which is assumed to depend only on the distance $d_{x,y}$ between them, that is, $\rho_{x,y} = \rho(d_{x,y})$. Under these assumptions, ecosystem invariability in a study area $A$ is (see Methods):

$$I(A) = \frac{1}{CV^2(A)} = I_1 \cdot \frac{A}{(A-1)\bar{\rho}_A + 1} \tag{1}$$

Here, $I_1 = 1/CV_1^2$ represents the invariability of a single patch, and $\bar{\rho}_A$ is the average temporal correlation between any two patches in area $A$.

Equation (1) shows that IAR is essentially governed by patterns of spatial synchrony across the landscape, which determines $\bar{\rho}_A$. Two limiting cases occur when the between-patch correlation is either 0 or 1 regardless of distance. In the absence of correlation (that is, $\rho_{x,y} = 0$), equation (1) becomes $I(A) = I_1 A$: invariability increases proportionally to area, and thus the slope of IAR on a log–log scale is 1 (Fig. 1). When patches are perfectly correlated (that is, $\rho_{x,y} = 1$), equation (1) becomes $I(A) = I_1$: invariability does not change with area because all patches fluctuate in a perfectly synchronous manner, and the slope of IAR is 0 (Fig. 1). In nature, however, spatial synchrony generally decreases with distance due to reduced environmental correlation, dispersal, and/or community similarity[24–27].

We consider two types of correlation–distance functions to predict IARs under more realistic scenarios. The first assumes an exponential decay ('light-tail') of correlation with distance: $\rho(d) = \rho_1 \times e^{-(d-1)/L}$, where $\rho_1$ represents local correlation, that is, the correlation between two neighbouring patches. $L$ is the characteristic correlation length beyond which correlation decreases steeply with distance, while $1/L$ measures the decay rate of correlation with distance. On a log–log scale, IARs then exhibit a triphasic curve, that is, invariability first increases steeply

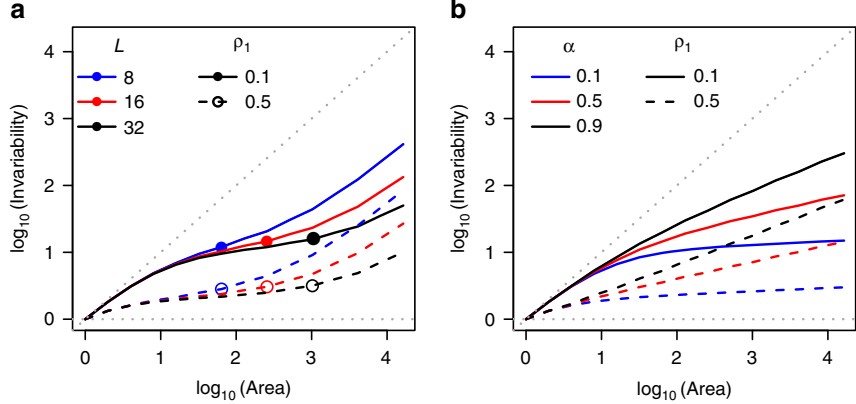

**Figure 1 | IARs in two-dimensional landscapes. (a,b)** IARs under exponential and power correlation–distance functions, respectively. $\rho_1$ represents the correlation between two neighbouring patches ($\rho_1 = 0.1$ in solid lines and 0.5 in dashed lines), $L$ is the characteristic length of the exponential decay ($L = 8$ in blue, 16 in red and 32 in black), and $\alpha$ represents the exponent of the power-law decay ($\alpha = 0.1$ in green, 0.5 in red and 0.9 in black). In **a**, the exponential decay yields triphasic IARs, that is, invariability increases steeply at small scales, more slowly at intermediate scales and steeply again at large scales (that is, beyond the area $L^2$, indicated by points/circles in **a**). In **b**, the power-law decay yields IARs that look more linear, especially from intermediate to large scales. Grey dotted lines show IARs under the two limiting cases in which between-patch dynamics are either perfectly correlated ($\rho_{x,y} = 1$ for all $x \neq y$; flat grey dotted lines) or independent ($\rho_{x,y} = 0$ for all $x \neq y$; beveled grey dotted lines).

with area, then increases more slowly, and eventually increases steeply again beyond an area around $L^2$ (Fig. 1a). The triphasic IAR can be understood as follows. Within the area $L^2$, between-patch correlation changes slightly with distance and stays at the magnitude of $\rho_1$; this relationship results in a relatively fast increase of invariability with area at the beginning, but the increase slows down and tends to saturate near $L^2$ (see Supplementary Note 1). Beyond $L^2$, between-patch correlation declines rapidly towards zero; invariability thus increases steeply with area, with a universal asymptotic slope of 1 (see Methods). The second function assumes a power-law decay ('heavy-tail'): $\rho(d) = \rho_1 \times d^{-\alpha}$, where $\alpha$ is the power-law exponent. In this case, IARs are quasilinear on a log–log scale (Fig. 1b). The initial slope is higher when $\rho_1$ is smaller, and the asymptotic slope converges to either $\alpha/2$ (when $\alpha < 2$) or 1 (when $\alpha \geq 2$) (see Methods).

Thus, the shape of IAR depends on the patterns of spatial synchrony across scales. On a log–log scale, the slope of IAR at area $A$ is determined by, and decreases with, the correlation between two neighbouring ecosystems with an area $A$ (see Methods). In particular, the initial slope of IAR decreases as local correlation ($\rho_1$) increases. The asymptotic slope of IAR increases with the exponent ($\alpha$) under the power-law correlation–distance function, and the pace of convergence to its universal asymptotic slope (that is, 1) increases with the decay rate ($1/L$) under the exponential function. Lastly, if, for any practical reason, one wants to describe IAR as a linear function on a log–log scale (as is often done for SAR[23]), the slope will decrease with local correlation ($\rho_1$) and increase with either the exponential decay rate ($1/L$) or the power-law exponent ($\alpha$) (Supplementary Fig. 1).

**Application of IAR to dataset on primary productivity.** We combined field[28,29] and remote sensing[30] data to derive IARs of primary productivity over the period 1985–2014. On a log–log scale, IARs exhibited triphasic curves from plot to continental scales (Fig. 2a). At the smallest scales (that is, below 1 km²), the invariability of primary productivity increased steeply with area at a decelerating rate. Beyond 1 km², in all five continents, invariability first increased slowly with area, and then increased steeply beyond $5 \times 10^5$ km². The slope varied more than 20-fold, between 0.03 and 0.8, depending on the spatial scale considered

(Table 1). The spatial synchrony of primary productivity generally decreased with distance (Fig. 2b,c). From intermediate to continental scales, spatial synchrony exhibited an exponential decay with distance, with estimates of $L$ in the range 570–792 km for four continents and 1,732 km for Australia (Table 1). As predicted by our model, the value of $L^2$ (that is, $3.3 \times 10^5 \sim 6.3 \times 10^5$ km² for the four continents) was consistent with the scale beyond which invariability increases steeply with area (that is, around $5 \times 10^5$ km²; see Fig. 2). Finally, note that spatial synchrony in the remote sensing data was much higher than that in the field data (Fig. 2b,c). This is because spatial synchrony depends not only on distance, but also on grain size. As grain size increases, spatial synchrony increases (see Supplementary Note 2).

**Application of IAR to dataset on North American bird biomass.** We also used data from the North American Breeding Bird Survey[31] to investigate IARs of total community biomass and individual species biomass. We selected 406 routes that had no missing records during the period 1990–2010 and that are located east of 100° W (see Methods). We used the number of survey routes as a surrogate for sampled area. The invariability of total community biomass increased quasilinearly with the number of routes, with a slope of 0.45 overall on a log–log scale. This slope, however, was larger at small scales ($z = 0.62$ from 1 to 32 routes) than at large scales ($z = 0.25$ from 32 to 406 routes) (Fig. 3a). The spatial synchrony of biomass fluctuations decayed as a power law with distance (Fig. 3b). We also derived species-level IARs for 121 bird species that were recorded at least twice during 1990–2010 in at least 100 routes. These IARs had slopes in the range $0.3 \sim 1$, with a mean value of 0.69 (Fig. 3). Interspecific variations in IAR slopes were related to those of spatial synchrony, as predicted by our model. Across species, IAR slopes decreased with the biomass correlation at 50 km (that is, the average distance of one route to its nearest neighbour) and increased with the power-law exponent of correlation decay (Fig. 3c,d).

The sampling scheme can potentially influence the calculation of invariability and hence the empirical patterns of IARs (see Supplementary Note 2). In the bird survey, although the hundreds of sampling routes cover the whole extent of eastern North America, the total sampling area represents only a relatively small proportion of the whole continent. Our

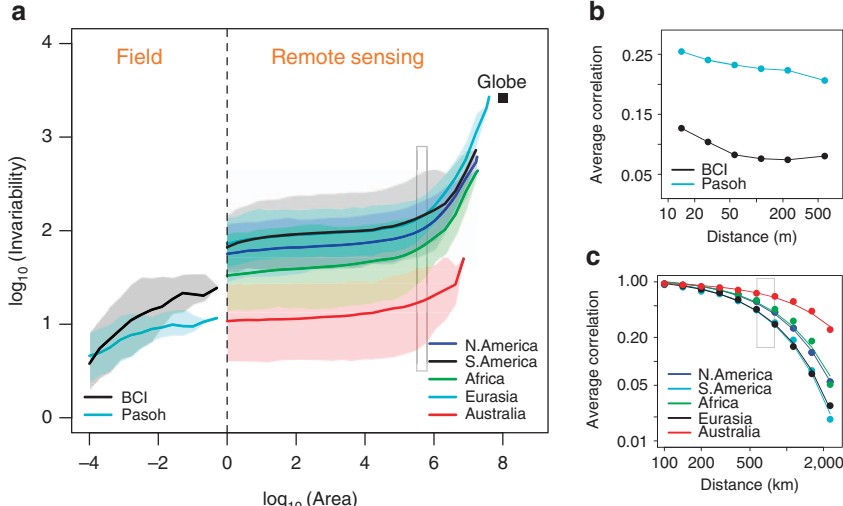

**Figure 2 | IARs and correlation–distance relationships of primary productivity.** (**a**) IARs of primary productivity from plot to continental scales (unit of area: km$^2$). Data are from field investigations of two 50-hectare tropical forest plots[28,29] (scale: $10^{-4}$–$0.5$ km$^2$) and remote sensing NPP products from MODIS[30] (scale: 1 km$^2$ to globe). For each plot or continent, the curve shows median invariability of 500 replicates, and shades show 25 and 75% quantiles. The square shows the invariability and area of the entire globe. The vertical dashed line corresponds to 1 km$^2$. (**b**) Correlation–distance relationships of primary productivity in the two tropical forest plots. Each point represents the average correlation between all subplot pairs (grid size: $10 \times 10$ m$^2$) corresponding to some distance category. (**c**) Correlation–distance relationships of primary productivity in the five continents based on MODIS data. Each point represents the average correlation between all grid pairs (grid size: $1° \times 1°$) corresponding to some distance category. Curves are fitted exponential functions: $\rho(d) = \rho_1 \times e^{-(d-1)/L}$ (see Table 1 for parameter values). The grey box in **c** indicates the range of the parameter $L$ for the four continents except Australia, which lies between 570 and 792 km. The grey box in **a** indicates the square of this range ($L^2$), that is, between $3.3 \times 10^5$ and $6.3 \times 10^5$ km$^2$.

**Table 1 | Correlation–distance relationships and IARs of primary productivity in five continents.**

| | Correlation–distance relationships $\rho(d) = \rho_1 \times e^{-(d-1)/L}$ | | | Invariability–area relationships (IARs) | | |
| --- | --- | --- | --- | --- | --- | --- |
| | $\rho_1$ | $L$ | $R^2$ | $z$ ($1 < A < 5 \times 10^5$ km$^2$) | $z$ ($A > 5 \times 10^5$ km$^2$) | $z$ ($A > 2 \times 10^6$ km$^2$) |
| North America | 1.00 | 7.51 | 0.998 | 0.032 | 0.503 | 0.596 |
| South America | 0.97 | 5.70 | 0.995 | 0.039 | 0.459 | 0.642 |
| Africa | 1.00 | 7.92 | 0.983 | 0.040 | 0.540 | 0.682 |
| Eurasia | 0.95 | 5.96 | 0.998 | 0.033 | 0.677 | 0.804 |
| Australia | 0.95 | 17.32 | 0.987 | 0.029 | 0.340 | 0.597 |

The relationships between average correlation ($\rho$) and distance ($d$) are modelled by an exponential function: $\rho(d) = \rho_1 \times e^{-(d-1)/L}$, where $\rho_1$ represents the correlation between two neighbouring grids and $L$ represents the characteristic correlation length. The unit of $d$ and $L$ is 100 km. For IARs, $z$ represents the log–log slope between invariability and area on respective scales.

theoretical model shows that incomplete spatial sampling could potentially increase the slope of IAR (see Supplementary Fig. 2). Thus, the reported slopes for community- and species-level IARs of bird biomass are likely to be higher than those based on full censuses; the latter, however, are impractical to obtain. The temporal sampling scheme may also influence the scaling patterns of invariability (see Supplementary Note 2)[32,33]. With the bird data, we examined how observation length might affect the slope of IAR. The results show that, over a shorter observation period (for example, 1990–2000), the IAR of bird community biomass had a slightly higher intercept and lower slope (Supplementary Fig. 3). The species-level IARs also exhibited higher intercepts over a shorter period, but no significant trend was detected for the slope (Supplementary Fig. 3).

## Discussion
Our theoretical and empirical analyses demonstrate a basic, yet fundamental, scaling pattern of ecological stability. While the spatial scaling of species diversity (that is, SAR) depends on the spatial distribution of species, that of invariability (that is, IAR) is mainly determined by the spatial synchrony of ecological

dynamics. The IARs of primary productivity exhibits triphasic curves, characterized by steeper increases in invariability at both small and large scales (Fig. 2). This relationship resembles the triphasic SARs for the world's flora and avifauna[34–37]. Although the remote sensing-based NPP estimates might be biased at small scales (for example, $\sim$ 1 km$^2$) (ref. 38), their reliability has been well validated at relatively larger scales (for example, > 50 km$^2$) (refs 39,40). Thus, uncertainty in NPP estimates should not affect the reported triphasic curve. The triphasic IARs may reflect the scale dependence of the determinants of spatial synchrony (see Supplementary Note 3). At the smallest scales, demographic stochasticity and observation error are likely to be prominent[41]. These factors are independent across space, which explains why spatial synchrony (between small grids, for example, 100 m$^2$) is low and invariability first increases steeply with area. At intermediate scales, the spatial correlation of environmental factors is likely to cause a relatively high spatial synchrony (between intermediate-size grids, for example, 1 km$^2$) (refs 24,25) and, consequently, a slower increase of invariability with area. At the largest scales, the spatial synchrony is low again due to a strong decorrelation of the environment and species composition

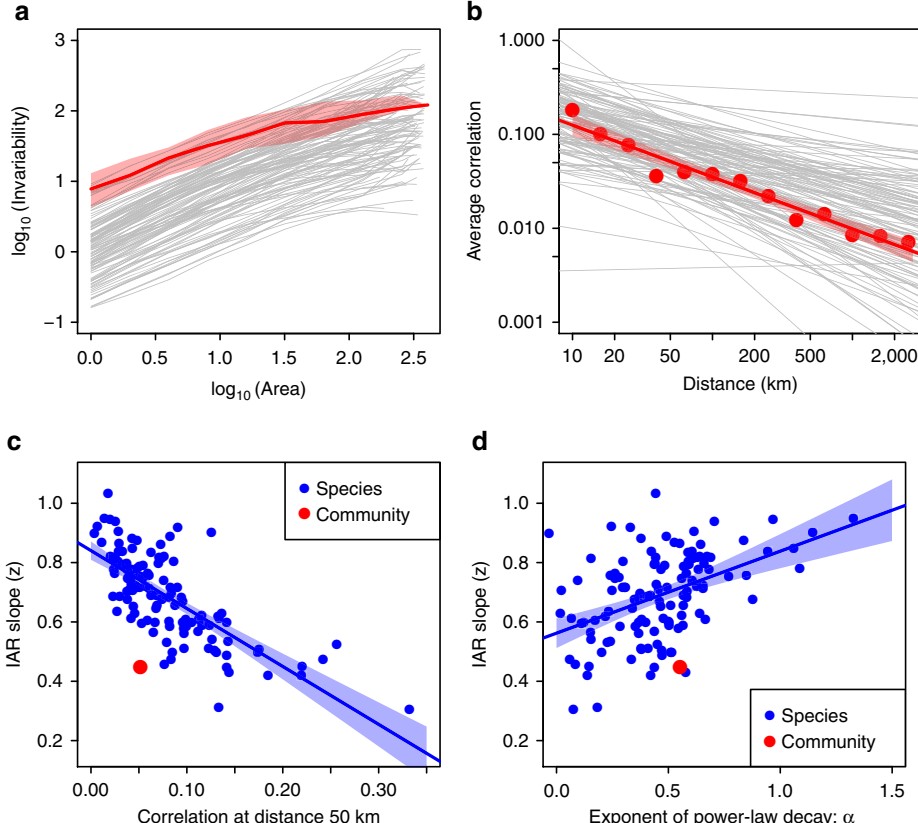

**Figure 3 | IARs and correlation–distance relationships of bird biomass.** (**a**) IARs of bird biomass across eastern North America[31]. Area is measured by the number of sampling routes (total number: 406). The red line shows median invariability of total community biomass across 406 replicates, and the red shade shows 25 and 75% quantiles. Grey curves show the IARs of individual species biomass for 121 bird species. (**b**) Correlation–distance relationships of bird biomass. The red points represent the average correlation of total community biomass corresponding to different distance categories. The red line is a fitted power function: $\rho(d) = 0.45 \times d^{-0.55}$ (unit of $d$: 1 km), with red shade representing 95% confidence intervals. Grey lines are fitted power functions for 121 bird species. (**c,d**) Relationship between the slope of species-level IARs and spatial correlation at a distance of 50 km (**c**) and the power-law exponent of correlation decay (**d**). The lines in **c,d** are least-square fits for species-level data, with blue shades representing 95% confidence intervals. The red points in **c,d** show values for community-level data.

beyond some correlation length (for example, $L$ in Table 1); this results in a steep increase of invariability with area. Furthermore, the upward inflection point of IARs of primary productivity (that is, $570 \sim 792$ km in four continents; see Table 1) is comparable to that of plant SARs (ca. 837 km)[36], implying that the correlation length of plant ecosystem dynamics is similar to that of the biogeography of the world's flora. Whether this parallel is due to ecological factors affecting both patterns, SAR affecting IAR, or mere coincidence remains to be explored.

In contrast, the spatial synchrony of bird biomass does not exhibit an accelerated decline at large scales, and the corresponding IARs do not exhibit the third stage of steep increase in invariability (Fig. 3). This might be due to the limited scale of the bird data we used. The correlation length of biogeographic processes of the world's avifauna, that is, the upward inflection point of its SAR, is about 1,585 km (ref. 36); this is comparable to the extent of our bird data (for example, 90% route pairs have a distance $< 1,900$ km). At a larger scale (for example, beyond North America), bird biomass might exhibit a rapid de-correlation and a steep increase in invariability.

The scale dependence of invariability has important implications for ecological research. Values of invariability need to be reported relative to the sample area (for example, quadrat size), just as species richness is relative to area. To investigate the effects of ecological factors (for example, species diversity) on invariability, one should compare sites with the same sampling scheme

and acknowledge the scale dependence of effect size, just as with biodiversity[42]. Moreover, IAR theory could be developed to generate the equivalent of rarefaction curves in order to compare studies with different sampling sizes, sampling intensities and time-series length[33,43,44].

IAR provides a novel tool for biodiversity conservation and ecosystem management. In particular, IARs could be used to predict the loss of population or ecosystem stability due to habitat destruction, in the same way as SARs have been used to predict species extinctions[45,46]. For instance, a power function $I(A) = cA^z$ could be used to predict changes in bird population invariability following habitat loss. Bird species exhibit large interspecific variations in their IARs, and disentangling their drivers would help us understand how different bird species may respond to habitat destruction differently. Similarly, IAR allows conservationists and managers to identify the minimum area needed to reduce uncertainty in the provision of ecosystem services below some threshold, for example, in agriculture and fishery[5–8], or the minimum area required for the long-term persistence of populations. Our study presented two examples of IARs on continuous and noncontinuous grids (that is, plant and avian communities, respectively). New studies are needed to generalize our results to other taxa and landscape structures, such as islands and island fragments, in order to apply IAR for real-world conservation and management[47]. Although large-scale spatio-temporal data are still very limited, this situation may

improve with the development of new techniques such as airborne light detection and ranging (LiDAR)[48,49].

IAR also has potential as a new tool to predict regime shifts (that is, Holling's resilience[21]) in spatial ecological systems. Spatial correlation has been used as an early warning signal for regime shifts in spatially structured ecosystems as correlations between neighbouring and distant patches are bound to increase prior to a shift[50]. These changes in spatial correlations in turn modify the shape of IAR qualitatively: in the proximity of a regime shift, the intercept of IAR, its initial and final slopes, all decrease so that the triphasic shape gradually diminishes (see Supplementary Fig. 4 and Supplementary Note 4). Furthermore, the decrease in the slope of IAR follows a specific pattern across scales. Spatial correlations propagate gradually through space, such that the initial slope of IAR decreases first, while the final slope decreases only close to the regime shift (Supplementary Note 4).

Our study provides an important first step towards understanding the spatial scaling of ecological stability. The findings here could serve as a benchmark for future research, and could be extended to compare the spatial scaling of different stability metrics and reveal their underlying mechanisms. While invariability is by far the most commonly used stability metric in empirical studies, theoretical research has mainly focused on asymptotic resilience[17]. Extending the IAR approach to investigate the spatial scaling of asymptotic resilience and other stability metrics[4,51] may contribute to developing multidimensional stability-area relationships, and thereby a more comprehensive understanding of the spatial scaling of stability. Future research also needs to explore spatial synchrony across scales quantitatively, in order to disentangle the drivers of IARs in theory and data. Such new body of research will contribute to transforming IAR into a practical tool for predicting the long-term responses of species and ecosystems to habitat and environmental changes.

## Methods

**Theoretical derivation of IAR.** We define ecosystem invariability in a study area A ($I(A)$) as the reciprocal of squared coefficient of variation ($CV^2$) of biomass in area A: $I(A) = 1/CV^2(A)$. Consider a two-dimensional landscape that consists of a grid of regularly distributed local patches (or ecosystems) of unit size. All local patches are assumed to have identical temporal mean ($\mu$) and variance ($\sigma^2$) of total biomass, that is, the mean and variance of time series of total biomass. We assume the temporal dynamics are stationary and hence $\sigma^2$ is constant through time, although empirical data may exhibit non-stationary dynamics. Between patches, ecosystem dynamics can be correlated with each other. The temporal correlation between patches x and y is denoted by $\rho_{x,y}$. Thus, the temporal variance of biomass in an area covering A patches is:

$$Var(A) = \sum_{x,y \in A} Cov(x,y) = \sigma^2 \cdot \left( A + \sum_{x,y \in A; x \neq y} \rho_{x,y} \right) = A\sigma^2(1 + (A-1)\bar{\rho}_A)$$

(2)

where $\bar{\rho}_A = \sum_{x,y \in A; x \neq y} \rho_{x,y} / [A(A-1)]$. The variability of total biomass in area A is thus:

$$CV^2(A) = \frac{Var(A)}{(A\mu)^2} = \frac{A\sigma^2(1 + (A-1)\bar{\rho}_A)}{(A\mu)^2} = \frac{1 + (A-1)\bar{\rho}_A}{AI_1}$$

(3)

where $I_1 = \mu^2/\sigma^2$. Therefore, ecosystem invariability in a study area A is (equation 1):

$$I(A) = \frac{1}{CV^2(A)} = I_1 \cdot \frac{A}{(A-1)\bar{\rho}_A + 1}$$

(1)

**Theoretical analyses of the slope of IAR.** On a log–log scale, the slope of IAR at area A ($z_A$) can be approximated by the change of invariability between areas A and 2A:

$$z_A = \frac{\log_2(I(2A)) - \log_2(I(A))}{\log_2(2A) - \log_2(A)} = \log_2(I(2A)) - \log_2(I(A))$$

(4)

By definition, $I(A) = (A\mu)^2/Var(A)$, and $I(2A) = (2A\mu)^2/Var(2A) = (2A\mu)^2/[2(1 + \rho_{AA}) \cdot Var(A)]$. Here, $\rho_{AA}$ represents the correlation between two

neighbouring patches with area A, which make up an area of 2A. Substitute $I(A)$ and $I(2A)$ into equation (4), we have:

$$z_A = \log_2 \frac{2}{1 + \rho_{AA}} = 1 - \log_2(1 + \rho_{AA})$$

(5)

Therefore, the slope of IAR at area A ($z_A$) decreases with the correlation between two neighbouring ecosystems both with area A. It is important to note that, whereas the correlation–distance functions in the main text (that is, $\rho(d)$) represent the correlation between two ecosystems with unit size, $\rho_{AA}$ denotes the correlation between two ecosystems with size A. So they are defined at different grain sizes, although the latter may be derived from the former. In the Supplementary Note 2, we show that given the distance, the correlation between two ecosystems increases with ecosystem size.

Below we investigate the initial and asymptotic slopes of IARs under the two correlation–distance functions considered in the main text. First, the initial slope ($z_{ini}$) is calculated by the log–log slope between $A = 1$ and $A = 2$. Following equation (5), we have: $z_{ini} = \log_2(2/(1 + \rho_1))$, where $\rho_1$ is local correlation, or the correlation between two neighbouring patches with unit area. So, the initial slope of IAR decreases with local correlation ($\rho_1$). Then, to derive the asymptotic slope ($z_{asym}$) of IAR, we calculate the log–log slope of IAR as the area A goes to infinity (see Supplementary Note 1):

$$z_{asym} = \frac{\ln I(A)}{\ln A} \cong -\frac{\ln(\bar{\rho}_A + A^{-1})}{\ln A}$$

(6)

In order to obtain $z_{asym}$, we need to derive the average pairwise correlation $\bar{\rho}_A$. Consider a square area $A = N^2$ (that is, a group of grids on $\{1, 2, …, N\} \times \{1, 2, …, N\}$), we have (see Supplementary Note 1):

$$\bar{\rho}_A = \frac{4}{N^2(N^2-1)} \sum_{k=1}^{N} \sum_{l=2}^{N} \left[ (N+1-k)(N+1-l) \cdot \rho(\sqrt{(k-1)^2 + (l-1)^2}) \right]$$

(7)

where $\rho(.)$ represents the correlation–distance relationship. By substituting the two correlation–distance functions into equation (7), we can derive $\bar{\rho}_A$ and $z_{asym}$ (see Supplementary Note 1). Under the exponential function, the asymptotic slope ($z_{asym}$) equals 1. Under the power-law function, we have:

$$z_{asym} = \begin{cases} \frac{\alpha}{2} & \alpha < 2 \\ 1 & \alpha \geq 2 \end{cases}$$

(8)

**Data sources.** Two data sets were used to study the IARs of primary productivity. The first included two 50-hectare tropical forest plots in Barro Colorado Island (BCI), Panama[28], and in Pasoh, Malaysia[29]. Both were established in 1980s and have since been surveyed 6–7 times. In each census, all woody stems with a diameter at breast height larger than 1 cm were identified, tagged and mapped. The aboveground biomass of each stem was estimated using an allometric equation[52]. Annual primary productivity was then estimated, for each of the $10 \times 10\,m^2$ subplots, as the total growth of biomass between two censuses divided by the length of the period (usually 5 years). The second data set consisted of MODIS-based remote sensing data of terrestrial net primary productivity (NPP) on a global scale[30]. For each grid ($0.00833° \times 0.00833°$), the annual NPP was estimated over the period 2000–2014. The estimation algorithm followed a radiation-use efficiency approach, and incorporated information on vegetation type and climate conditions[30]. This approach provides the first operational, near-real-time calculation of global NPP[39]. The MODIS NPP product is one of the most reliable estimates of vegetation dynamics at the global scale[30]. Its reliability has been validated across scales with eddy flux-based estimations and field measurements[38–40,53].

We also used the data set from the North American Breeding Bird Survey[31] to study the IARs of bird biomass. We selected routes without missing records between 1990 and 2010; among these 555 routes, we used the 406 routes that are located east of 100° W. This is because sampling density was much higher east of 100° W (average distance of one route to its nearest neighbour: ~ 50 km) than west (average distance: ~ 100 km) (Supplementary Fig. 5), and the 100° W parallel is also the location of sharp gradients of annual precipitation[54]. For each route, the population size of each encountered species was recorded and multiplied by its mean body mass[55] to estimate the species biomass. Total community biomass was obtained by summing up the biomass of all species.

**Empirical IARs and correlation–distance relationships.** To construct the IARs of aboveground primary productivity in BCI and Pasoh, we first randomly selected a starting grid (that is, a $10 \times 10\,m^2$ subplot), and then increased the area to $200\,m^2$, $400\,m^2$, …, $5 \times 10^6\,m^2$, by including the neighbouring grids. The procedure was repeated 500 times. For the satellite-based NPP data, IARs were constructed for each of the five continents, with similar procedures as in BCI and Pasoh (Supplementary Fig. 6). Because the original data were based on a resolution of 0.00833 degree, surface area differed between grids at high and low latitudes. We thus converted the longitude-latitude distance to Euclidean distance, and calculated the area for each grid.

The IARs of bird biomass were constructed similarly as above. However, the area in this case was not real area, but the sampled area or the number of routes. We first calculated the IAR of total community biomass. By taking different routes as the starting route, we obtained 406 replicates. In each replicate, we started with one route and increased the number of routes to 2, 4, 8, …, 256, 406 by including the closest routes. We also derived IARs for 121 bird species that were recorded at least twice during 1990–2010 in at least 100 routes, with similar procedures as for total community biomass.

For both data sets, we calculated the temporal correlation for all pairs of grids (for plant data) or routes (for bird data). We grouped these pairs (of grids or routes) into different categories according to their distance, and computed the average correlation for each distance category. For computational reasons, the calculation for the Modis NPP data was based on a grid size of $1^o \times 1^o$; thus, the distance between two neighbouring grid was about 100 km around the equator and smaller around the poles. Note that the calculation for the two tropical forest plots was based on a grid size of $10 \times 10\,m^2$, much smaller than that for the Modis-NPP data. In Supplementary Note 2, we showed that, given the distance, spatial synchrony increased with grid size. This explained why the spatial synchrony of the Modis NPP data was much larger than that of the forest plot data (Fig. 2b,c).

**Data availability.** The data that support the findings of this study are available on request from the BCI Forest Dynamics Plot data set (http://ctfs.si.edu/webatlas/datasets/bci/), the Pasoh Forest Dynamics Plot data set (http://www.ctfs.si.edu/site/pasoh/), the MODIS GPP/NPP Project (MOD17) (http://www.ntsg.umt.edu/project/mod17) and the North American Breeding Bird Survey (https://www.pwrc.usgs.gov/bbs/).

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

## Acknowledgements

We thank J. Chave, J. Clobert, W. Deng, B. Haegeman, A.-S. Lafuite, P. Legendre, C. Yue and Y.R. Zelnik for helpful discussions, and U. Brose, J. Chase, A. Gonzalez,

B. Haegeman and J. Kolasa for comments on earlier versions of the manuscript. We also thank the Smithsonian Tropical Research Institute for making the BCI data sets publicly available and the thousands of U.S. and Canadian participants who annually perform and coordinate the North American Breeding Bird Survey. This work was supported by the TULIP Laboratory of Excellence (ANR-10-LABX-41) and by the BIOSTASES Advanced Grant, funded by the European Research Council under the European Union's Horizon 2020 research and innovation programme (grant agreement No 666971). S.W. gratefully acknowledges the support of the German Centre for Integrative Biodiversity Research (iDiv) Halle-Jena-Leipzig funded by the German Research Foundation (FZT 118).

## Author contributions

S.W., M.L. and C.d.M. designed the project; S.W. developed the model and analysed the data, with input from J.-F.A., J.F., K.A.R. and S.T.; J.-F.A. developed the metapopulation model in Supplementary Note 4; S.W. wrote the first draft of the manuscript, and M.L., J.-F.A. and C.d.M. contributed to the revision.

## Additional information

**Competing interests:** The authors declare no competing financial interests.

