## [Peer Review File · Nature Communications]

Reviewers' Comments:

Reviewer #1 (Remarks to the Author):

The goal in Science is to translate careful observations into understanding. In my field, few times I have had the chance to witness how new observational patterns are found, described, and so nicely reported. The authors have all the merit to achieve this goal.

This paper introduces a novel spatial pattern in Ecology. The authors investigate how different measures of ecosystem stability change with observational scale. They show that the shape and slope of this scaling, which they named the Stability Area Relationship (StAR), are largely determined by the patterns of spatial synchrony of ecological dynamics. The authors check their ideas using two types of data. On the one hand, data on biological primary productivity from local up to continental scales. On the other hand, spatio-temporal community data on North American Birds.

Below you will find some comments that may enrich and clarify certain aspects of the work. I hope the authors will find them useful.

Aggregation always tends to dampen down variability. Whether you look at animal counts in Serengeti, you measure the energy of a number of molecules in a container, or the number of phytoplankton cells per water volume, increasing observation scale will always decrease variability. Therefore, the increasing nature of a StAR curves is not surprising at all. Damped variability (increasing stability) results from aggregation of partially-decoupled sub-units of any spatially extended system as the aggregation scale increases. This is quite general. The authors show that the underlying degree of spatial coupling, which is measured in terms of spatial correlation, determines the shape of StARs. So, on one hand, we have the observational pattern. Whenever we have good spatio-temporal data, we will be able to calculate nice StAR curves. But, on the other hand, we have the underlying mechanisms producing that pattern. In fact, we are always interested in the latter, which, in this case, are the underlying causes of variability and spatial synchrony. Questions such as to what extent environmental variability rather than population dispersal cause population synchrony rather than spatial decoupling (or the other way around) are the ones ecologists are mostly interested in. The authors discuss clearly how a pattern of spatial correlation determines StARs curves, but spend less time in discussing about the implication in the opposite direction. Can the same shape of a StAR curve be produced by contrasting different mechanisms? In other words, what is the discrimination power of this

pattern when used to disentangle underlying ecological dynamics?

The authors claim that this type of relationships provide a quantitative tool to predict the effects of habitat loss on population and ecosystem stability. I have the feeling that the link between their theoretical findings and conservation biology can be better substantiated. As a proof of concept, can the authors build an example showing how habitat loss (area reduction) through lessening ecosystem stability induces population extinction? Keep in mind that a type of habitat loss, i.e., habitat fragmentation through increasing population decoupling could also reduce overall extinction risk. A pedagogical example would only contribute to make their contribution even stronger.

In principle, different descriptors of the dynamics of the same ecosystem (for instance, species richness and total biomass per unit area) may show different StAR curves. If this is correct, we cannot talk about the stability of a whole ecosystem per se, but instead about the stability of a certain ecosystem property. Can this be interpreted as a clear weakness of your approach?

In physical systems, correlation length reaches the size of the whole system close to critical points. In Ecology, critical transitions are particular difficult to predict. Some early warning signals are based on spatial correlation. In a spatially extended systems, what would be the signature on the StAR curve of proximity to a regime shift? Could StAR be used to predict regime shifts?

To sum up, overall the paper reads well and supplementary material is easy to follow.

Reviewer #2 (Remarks to the Author):

The authors introduce a new concept for estimating how ecosystem stability is dependent on the spatial scaling: Stability – Area Relationship StAR. They present the theoretical derivation of the concept, perform theoretical analyses and also test it on two different empirical data sets. The model predicts that StARs are largely determined by patterns of spatial correlation of ecological dynamics (such as spatial correlation in species biomasses).

In general I think this is an interesting paper, introducing a novel concept that potentially could be of use both in theoretical studies and in the extension perhaps more applied research. That said the paper is quite complex and compact. In some parts the descriptions are hard to follow and it would be valuable if the authors could add some extra effort in details that would increase readability.

For example, I would like to see a developed introduction section, where the importance and usefulness of the StARs is more elaborated. The whole paper is as it stands very technical, and in a journal as Nature Communication it is nice if more accessible to a wider audience. Yes, I am aware of the restricted number of words, but just some carefulness with the language and setting the scene to show that this IS exciting could have a large impact.

In general it would be good if the authors had a look at the wording used for describing different factors. One example is that “Forest plot” and “Mondis NPP” is used in figure 2, but not in the main text.

Another example is that both $1/L$ and α is used to describe the same quantity (decay in correlation with distance).

In Figure 1 the value of L is printed out in the figure, but what is interesting is the rate of decay with distance ($1/L$). As it is the decay that is referred to in the text when describing the different relationships it would be better having this highlighted in the figure. In general, the figure text for Figure 1 needs to be extended, as it cannot be understood now without careful reading of the main text.

The paper is as said complex so all these small considerations is valuable for the reader.

Other minor comments

l. 126 You refer to BIC data without stating earlier what that is (it is stated in the methods only).

l. 227 “... is determined by the correlation between...” – specify what it is that is correlated. This explanation is in general hard to follow and in some places there are contradictive statements. Please clarify.

In the description of the methods (p 11) state the definition of $S(A)$ (ecosystem stability in a study area A , which is stated in the “main text”, but should also be stated here.

Reviewer #3 (Remarks to the Author):

NCOMMS-16-19499

Spatial scaling of ecological stability: the Stability-Area Relationship

by

Shaopeng Wang, Michel Loreau, Jingyun Fang, Jean-Francois Arnoldi, K. Abd. Rahman,

Shengli Tao, Claire de Mazancourt

This paper sets out to introduce to ecology and promote a new quantity called the Stability Area Relation. The authors define this quantity, derive its behaviour under various assumptions and evaluate its magnitude for global NPP and the biomass of N. American birds.

This paper is a step in the right direction. We need more investigation of spatio-temporal variability in ecological systems. However, whether the StAR (as defined here) is up to the task is not fully established in this paper. I found no mistakes in the maths but definitions and explanations are sloppy in places.

General remarks for the authors

How original is the StAR? It shares properties of the more intuitively direct variogram /semivariogram used in spatial statistics (Cressie 1993). The authors should be specific in their argument for launching a new S[x]R into ecology - why is the variogram not enough? According to the authors "Our theoretical and empirical analyses demonstrate a basic, yet fundamental, scaling pattern of ecological stability" but from a myriad of possible ecological variables, just two examples are used, one that agrees with their model the other maybe. Such empirical support is OK as a proof of concept or a pilot study. But more support is needed.

I am also concerned about the loose definition and analysis in places. Firstly, the StAR should be defined more carefully via a formula. The current loose definition is not sufficient. Secondly, greater care should be taken in their explanation of multi-scale variability, where scale mismatches are a common source of error. Even the simplest spatio-temporal process has sampling-dependent parameters. These include the following:

1. Upper resolution scale. This is A .
2. Lower scale of resolution. Set to unity in this paper.
3. Inter-patch distance. Set to one in this paper (grid is flush in model)
4. Time-sampling behavior. In time too, ecological variables are multi-scale stochastic processes. Again there is a lower scale, an upper scale and a sampling rate. (Halley 2007).

Each of the above will affect how variability scales and hence the structure of the StAR. I found the explanation of these very scattered. The authors have clearly thought about the issues but they need to tell readers in a more structured way. The temporal issues in particular are skirted over very quickly. "temporal variance" is used (in Line 74) but not defined precisely. Stationarity is assumed in line-73, though the authors are aware that variance grows with observation time.

Finally, why is the word "stability" chosen, rather than variability? Stability is typically associated with dynamical systems, where it is an internal property of the ecological community. However, the StAR is a measure of variability that may simply be a linear response to externally imposed abiotic variability.

Specific issues:

L-35. Ref Pimm (1991) or Pimm & Redfearn (1988). Important contribution to the dynamic

nature of ecological variability.

L-91. The exponential model used is confusing due to the use of l(one) in one case and l(ell) in another. Which is it? Maybe an alternative lowercase letter for the grid address in Methods might be preferable.

L-192. "StARs could be used to predict the loss of population or ecosystem stability due to habitat destruction, in the same way as SARs". This claim also needs to be justified better.

Lines 214-224. This section is confusing. Most of the problems flow from the failure to define S(A) precisely.

References

1. Cressie, Noel. "Statistics for spatial data: Wiley series in probability and statistics." Wiley-Interscience, New York 15 (1993): 105-209.
2. Halley, John M. "How do Scale and Sampling Resolution Affect Perceived Ecological Variability and Redness?" in *The Impact of Environmental Variability on Ecological Systems*. Ed. D.A. Vasseur and K.S. McCann, Springer, Netherlands, 2007. 17-40.
3. Pimm, Stuart L. *The balance of nature?: ecological issues in the conservation of species and communities*. University of Chicago Press, 1991.
4. Pimm, Stuart L., and Andrew Redfearn. "The variability of population densities." *Nature* 334 (1988): 613-614.

Reviewer #4 (Remarks to the Author):

This manuscript proposes a measure of spatial stability for ecological systems. The manuscript then goes on to investigate various aspects of this measure both empirically and theoretically. I will separate my comments into two parts with the first focusing on what are essentially matters of presentation and the second on the underlying concepts.

I have real concerns with the presentation because I do not believe that the measure presented is a stability measure. Variability and stability are difference concepts. Stability should only relate (and is defined as) abilities to withstand perturbations and either remain close to some state (or return to the state). The variability of the system is determined by both the stability and the perturbations, and equating stability and variability (or more precisely the inverse of variability) is a very poor idea and is liable to lead to confusion. Thus, I would strongly argue against publication of the paper as is, leading to the question of whether something interesting and novel is contained in the underlying ideas of spatial variability as developed in the manuscript.

Could an interesting paper be developed by instead focusing on an idea like spatial patterns of variability? I am also not convinced that this is possible. As far as I can tell, the theoretical development is simply a presentation of different patterns of spatial variability – the idea that variability relates to synchrony does not provide insights. Perhaps the most novel part of the paper is the empirical analysis of the spatial patterns of variability for different taxa. However, this likely can be simply explained by different dispersal distances.

Reviewers' comments:

Reviewer #1 (Remarks to the Author):

The goal in Science is to translate careful observations into understanding. In my field, few times I have had the chance to witness how new observational patterns are found, described, and so nicely reported. The authors have all the merit to achieve this goal.

This paper introduces a novel spatial pattern in Ecology. The authors investigate how different measures of ecosystem stability change with observational scale. They show that the shape and slope of this scaling, which they named the Stability Area Relationship (StAR), are largely determined by the patterns of spatial synchrony of ecological dynamics. The authors check their ideas using two types of data. On the one hand, data on biological primary productivity from local up to continental scales. On the other hand, spatio-temporal community data on North American Birds.

Below you will find some comments that may enrich and clarify certain aspects of the work. I hope the authors will find them useful.

Response: Thank you for your comments. We are glad that the reviewer values our contribution.

Aggregation always tends to dampen down variability. Whether you look at animal counts in Serengeti, you measure the energy of a number of molecules in a container, or the number of phytoplankton cells per water volume, increasing observation scale will always decrease variability. Therefore, the increasing nature of a StAR curves is not surprising at all. Damped variability (increasing stability) results from aggregation of partially-decoupled sub-units of any spatially extended system as the aggregation scale increases. This is quite general. The authors show that the underlying degree of spatial coupling, which is measured in terms of spatial correlation, determines the shape of StARs. So, on one hand, we have the observational pattern. Whenever we have good spatio-temporal data, we will be able to calculate nice StAR curves. But, on the other hand, we have the underlying mechanisms producing that pattern. In fact, we are always interested in the latter, which, in this case, are the underlying causes of variability and spatial synchrony. Questions such as to what extend environmental variability rather than population dispersal cause population synchrony rather than spatial decoupling (or the other way around) are the ones ecologists are mostly interested in. The authors discuss clearly how a pattern of spatial correlation determines StARs curves, but spend less time in discussing about the implication in the opposite direction. Can the same shape of a StAR curve be produced by contrasting different mechanisms? In other words, what is the discrimination power of this pattern when used to disentangle underlying ecological dynamics?

Response: It is true that an increasing StAR is not surprising. But this does not make it uninteresting. The species-area relationship (SAR) is not surprising either; yet, over the past century ecologists have devoted much effort to explaining “why species-area relationships show strong and recurrent qualitative and quantitative patterns” (Hubbell 2001) and using it as a quantitative tool to predict biodiversity changes (Lomolino 2000). Similar arguments apply to StARs. The objective of our paper is to introduce the concept of StAR and study its basic properties quantitatively, e.g. how spatial synchrony affects the shape and slope of StAR and how StARs fits empirical data. Our work is an important first step towards understanding the spatial scaling of stability. Our hope is that our paper will stimulate new research

efforts to collect new data across taxa and landscape configurations to reveal the empirical shape of StARs, develop new theories to clarify how StAR emerges from lower-level ecological processes, and finally transform the concept of StAR into a practical tool for predicting the impacts of habitat changes on ecological stability. In the revised manuscript, we have revised and added several sentences in the concluding paragraph to clarify this. Please refer to Page 11 Lines 246-254.

The reviewer raises an interesting point concerning the discrimination power of StARs in disentangling underlying ecological processes. It is obviously exciting to be able to infer processes from patterns; however, this is generally difficult in ecology. Ecosystems are complex systems, and patterns observed at one scale are often regulated by processes operating at multiple scales (Levin 1992). This makes it difficult to match patterns and processes. Indeed, to our knowledge, few patterns in ecology can distinguish between alternative processes in a convincing manner. For instance, the latitudinal biodiversity gradient has historically stimulated hundreds of hypotheses, but without a consensus till now. Another example is the recent debate on using species abundance distributions to disentangle niche vs. neutral processes (Chisholm & Pacala 2010). Similarly, the scaling patterns of stability (StAR) and biodiversity (SAR), which are also shaped by multiple processes, might not be sufficient to disentangle underlying processes.

This being said, future research may develop new quantitative approaches to explore how different mechanisms may generate different patterns of spatial synchrony and StARs. In the Supplementary Note 3, we have summarized several potentially important mechanisms, including environmental correlation, dispersal, species distribution, and others (Lande et al. 1999; Liebhold et al. 2004; Wang & Loreau 2016). Efforts in this direction may provide some hints on how to disentangle drivers of StARs from data (e.g. novel experimental designs and data collection protocols). In the revised manuscript, we have added one sentence in the concluding paragraph to clarify this: “*Future research also needs to explore spatial synchrony across scales quantitatively, in order to disentangle the drivers of StARs in theory and data*” (Page 12 Lines 251-252).

The authors claim that this type of relationships provide a quantitative tool to predict the effects of habitat loss on population and ecosystem stability. I have the feeling that the link between their theoretical findings and conservation biology can be better substantiated. As a proof of concept, can the authors build an example showing how habitat loss (area reduction) through lessening ecosystem stability induces population extinction? Keep in mind that a type of habitat loss, i.e., habitat fragmentation through increasing population decoupling could also reduce overall extinction risk. A pedagogical example would only contribute to make their contribution even stronger.

Response: We are happy to see that the reviewer envisages this exciting direction for StARs. One of the major strengths of StAR is that it may provide a novel tool to predict the effects of habitat loss on ecological stability, just as SAR does in biodiversity conservation. This implication has been highlighted in the Discussion of our manuscript. In the revised manuscript, we have added two sentences to better explain this: “*For instance, a power function $S(A)=cA^z$ could be used to predict changes in bird population stability following habitat loss. Bird species exhibit large interspecific variations in their StARs, and disentangling their drivers would help us understand how different bird species may respond to habitat destruction differently*” (Page 10 Lines 224-227).

The reviewer suggests to further clarify this implication with a concrete example. We fully agree with the value of doing this, but we believe that a new significant project is required to achieve this in a convincing way. On the one hand, habitat loss and fragmentation can result from different scenarios with

different consequences on stability (just as for SAR: see Hanski et al. 2013). On the other hand, species traits (e.g. dispersal characteristics) can interact with landscape changes and complicate predictions. In order to apply StARs to real-world conservation and management, these complexities should be incorporated in more specific models. For instance, our ongoing work is exploring how species' body mass and trophic level may affect their StARs and thus their responses to habitat destruction. Therefore we prefer to keep these extensions for future work.

In principle, different descriptors of the dynamics of the same ecosystem (for instance, species richness and total biomass per unit area) may show different StAR curves. If this is correct, we cannot talk about the stability of a whole ecosystem per se, but instead about the stability of a certain ecosystem property. Can this be interpreted as a clear weakness of your approach?

Response: We fully agree that different StAR curves can be developed for the same ecosystem. But, instead of considering it a weakness, we consider it a strength as it makes this tool flexible and applicable to a wide range of ecological properties. An exciting future direction would be to develop a multidimensional approach to StAR.

Stability quantifies the ability of a system to withstand perturbations. To account for different types of perturbations and different aspects of ecosystem responses, stability has historically been developed into a multidimensional concept, which includes such different components as resistance, resilience, persistence, variability, etc. (Donohue et al. 2013). Multidimensional approaches to ecological stability are important to develop a full understanding of ecosystem responses in the face of anthropogenic and natural perturbations (Barros et al. 2016; Donohue et al. 2016). To this end, any single stability metric might not be sufficient to represent “the stability of a whole ecosystem per se”. Thus, a multidimensional StAR could be developed by studying the spatial scaling of multiple stability metrics. In the revised manuscript, we added one sentence in the concluding paragraph to mention this future direction: “*As stability can be defined in multiple dimensions, a multidimensional StAR could be developed to achieve a more comprehensive understanding of spatial scaling of stability*” (Page 11 Lines 249-251).

In physical systems, correlation length reaches the size of the whole system close to critical points. In Ecology, critical transitions are particular difficult to predict. Some early warning signals are based on spatial correlation. In a spatially extended systems, what would be the signature on the StAR curve of proximity to a regime shift? Could StAR be used to predict regime shifts?

Response: Thank you for this insightful comment. Our paper shows an explicit link between spatial correlation and StAR; therefore, StAR could serve as a tool to predict regime shifts, just as spatial correlation does (Dakos et al. 2010). To demonstrate this, we have developed a dynamical metpopulation model in which the metpopulation has two alternative stable states (either an underexploited state with high biomass or an overexploited state with low biomass) depending on the harvesting rate. Our model shows that, as the metpopulation approaches the regime shift, the intercept of StAR, its initial and final slopes, all decrease so that the triphasic shape gradually diminishes. Furthermore, the decrease in the slope of StAR follows a specific pattern across scales. Spatial correlations propagate gradually through space, such that the initial slope of StAR decreases first, while the final slope decreases only close to the regime shift. Thus, StAR has great potential as a new tool to predict regime shifts in spatial ecological systems. In the revised manuscript, we have added a new paragraph in the discussion to illustrate this

implications (Page 11 Lines 236-245). We have also added a new supplementary note (i.e. Supplementary Note 4) to explain the details of our metapopulation model.

To sum up, overall the paper reads well and supplementary material is easy to follow.

Response: Thank you for your enthusiastic and inspiring comments.

Reviewer #2 (Remarks to the Author):

The authors introduce a new concept for estimating how ecosystem stability is dependent on the spatial scaling: Stability – Area Relationship StAR. They present the theoretical derivation of the concept, perform theoretical analyses and also test it on two different empirical data sets. The model predicts that StARs are largely determined by patterns of spatial correlation of ecological dynamics (such as spatial correlation in species biomasses).

In general I think this is an interesting paper, introducing a novel concept that potentially could be of use both in theoretical studies and in the extension perhaps more applied research. That said the paper is quite complex and compact. In some parts the descriptions are hard to follow and it would be valuable if the authors could add some extra effort in details that would increase readability.

Response: Thank you for your comments. We have revised the manuscript throughout to improve readability.

For example, I would like to see a developed introduction section, where the importance and usefulness of the StARs is more elaborated. The whole paper is as it stands very technical, and in a journal as Nature Communication it is nice if more accessible to a wider audience. Yes, I am aware of the restricted number of words, but just some carefulness with the language and setting the scene to show that this IS exciting could have a large impact.

Response: In the revised manuscript, we have added several sentences in the Introduction to better clarify the importance and usefulness of StARs for a wider audience: “*Such a variability-based stability metric measures the temporal constancy in the functioning of ecosystems or in the size of populations, and hence in the ecosystem services they deliver to human societies. The spatial scaling of variability thus informs us about how the reliability of ecosystem services may change across scales*” (Page 4 Lines 59-62), and “*We hope that StAR will open new research prospects similar to the classical Species-Area Relationship (SAR), with potentially as wide and important applications*” (Page 5 Lines 86-88).

We have also extended the Introduction in two other directions. First, we added several sentences explaining that there is a continuity in thinking about stability-scale relationships between our paper and a few previous papers (Peterson et al. 1998; Jorgensen & Nielsen 2013; Wang & Loreau 2014). These papers hinted at the idea that stability might change with scale. Our StAR approach is consistent with these early thoughts, but it provides a more quantitative framework to study the spatial scaling of stability. Please refer to Page 3 Lines 43-53. Second, we added several sentences to justify the use of variability-based stability metrics. For instance, we explain that variability-based metrics reflect the interplay

between intrinsic dynamical processes and perturbations that act upon a system, and it can be consistently defined across organization and scales. Please refer to Page 4 Lines 64-72.

In general it would be good if the authors had a look at the wording used for describing different factors. One example is that “Forest plot” and “Modis NPP” is used in figure 2, but not in the main text.

Response: In the revised manuscript, we have replaced “Forest plot” and “Modis NPP” in Figure 2 by “Field” and “Remote sensing”, respectively. We have also revised the figure caption accordingly to make it consistent: “*Data are from field investigations of two 50-hectare tropical forest plots (scale: 10-4 to 0.5 km²) and remote sensing NPP products from MODIS (scale: 1 km² to globe)*”. Please refer to Page 25 Lines 504-506.

Another example is that both $1/L$ and α is used to describe the same quantity (decay in correlation with distance).

Response: Sorry for this confusion that was introduced by our referring to both parameters ($1/L$ and α) as “decay rate”. There exists a fundamental difference between exponential and power-law functions; due to this, the two parameters ($1/L$ and α) are not replaceable. For the exponential decay, there is a characteristic scale, L , beyond which correlation drops off to zero quickly, and $1/L$ measures the decay rate of correlation. For the power-law decay, α is the power-law exponent, instead of a rate that has a unit of km^{-1} or year^{-1} . In the revised manuscript, we have corrected this and referred to “ α ” always by “*the power law exponent*” (Page 6 Line 125). We have also added some words to better clarify L : “*L is the characteristic correlation length beyond which correlation decreases steeply with distance, while $1/L$ measures the decay rate of correlation with distance*” (Page 6 Lines 115-116).

In Figure 1 the value of L is printed out in the figure, but what is interesting is the rate of decay with distance ($1/L$). As it is the decay that is referred to in the text when describing the different relationships it would be better having this highlighted in the figure. In general, the figure text for Figure 1 needs to be extended, as it cannot be understood now without careful reading of the main text.

Response: The parameter L has its own important meaning: (i) it is the characteristic length beyond which correlation decreases steeply with distance; (ii) its square indicates the upward inflection point beyond which stability increases steeply with area (Page 6 Lines 115-119). In figure 1, we have highlighted the area L^2 along StAR curves to indicate the inflection point. So we prefer to highlight L , instead of $1/L$. In the revised manuscript, we have added several sentences in the caption of Figure 1 to clarify this as well as make it self-explanatory: “*L is the characteristic length of the exponential decay, and α represent the exponent of the power-law decay. In (a), the exponential decay yields triphasic StARs, i.e. stability increases steeply at small scales, more slowly at intermediate scales, and steeply again at large scales (i.e. beyond the area L^2 , indicated by points/circles in a). In (b), the power-law decay yields StARs that look more linear, especially from intermediate to large scales*” (Page 24 Lines 493-497).

The paper is as said complex so all these small considerations is valuable for the reader.

Response: Thank you. We have revised the manuscript following your suggestions.

Other minor comments

l. 126 You refer to BIC data without stating earlier what that is (it is stated in the methods only).

Response: In the revised manuscript, we have replaced “BCI data” by “field data”.

l. 227 “... is determined by the correlation between...” – specify what it is that is correlated. This explanation is in general hard to follow and in some places there are contradictive statements. Please clarify.

Response: Sorry for the confusion. In the original manuscript, this argument was supported by the calculation following it. For clarity, in the revised manuscript we have moved this sentence after the calculation and revised it to be more specific: “*the slope of StAR at area A (z_A) decreases with the correlation between two neighboring ecosystems both with area A*” (Page 13 Lines 280-281).

In the description of the methods (p 11) state the definition of $S(A)$ (ecosystem stability in a study area A, which is stated in the “main text”, but should also be stated here.

Response: In the revised manuscript, we have added some words to clarify the definition of $S(A)$ in the Methods: “*We define ecosystem stability in a study area A ($S(A)$) as the reciprocal of squared coefficient of variation (CV^2) of biomass in area A: $S(A)=1/(CV^2 (A))$ ” (Page 12 Lines 258-259).*

Reviewer #3 (Remarks to the Author):

NCOMMS-16-19499

Spatial scaling of ecological stability: the Stability-Area Relationship

by

Shaopeng Wang, Michel Loreau, Jingyun Fang, Jean-Francois Arnoldi, K. Abd. Rahman, Shengli Tao, Claire de Mazancourt

This paper sets out to introduce to ecology and promote a new quantity called the Stability Area Relation. The authors define this quantity, derive its behaviour under various assumptions and evaluate its magnitude for global NPP and the biomass of N. American birds.

This paper is a step in the right direction. We need more investigation of spatio-temporal variability in ecological systems. However, whether the StAR (as defined here) is up to the task is not fully established in this paper. I found no mistakes in the maths but definitions and explanations are sloppy in places.

Response: Thank you for your comments. The objective of the paper is to introduce the concept of StAR as a new approach to study the spatial scaling of stability and variability. We agree that our paper does not offer a complete solution to understanding the spatial scaling of stability. Indeed, we doubt that this can be done in a single scientific paper. By introducing the concept of StAR and investigating its theoretical and empirical properties, our paper provides an important first step towards this goal. Our hope is that our study will stimulate new research efforts to collect new data across taxa and landscape configurations to

reveal the empirical shape of StARs, develop new theories to clarify how StAR emerges from lower-level ecological processes, and finally transform the concept of StAR into a practical tool for predicting the impacts of habitat changes on ecological stability. In the revised manuscript, we have revised the concluding paragraph to clarify this. Please refer to Page 11 Lines 246-254.

Besides, we have revised the manuscript following your other suggestions. Please see our responses below.

General remarks for the authors

How original is the StAR? It shares properties of the more intuitively direct variogram /semivariogram used in spatial statistics (Cressie 1993). The authors should be specific in their argument for launching a new S[x]R into ecology - why is the variogram not enough? According to the authors “Our theoretical and empirical analyses demonstrate a basic, yet fundamental, scaling pattern of ecological stability” but from a myriad of possible ecological variables, just two examples are used, one that agrees with their model the other maybe. Such empirical support is OK as a proof of concept or a pilot study. But more support is needed.

Response: In spatial statistics, a variogram depicts the pattern of spatial dependence of some focal variable (e.g. gold percentage). It is commonly represented as a graph that shows the variance or (squared) difference in measure with distance between sample locations (Cressie 1993). Due to spatial autocorrelation, the difference or variance between two samples generally increases with distance, and the correlation between two samples decreases with distance. The latter pattern is conceptually related to the spatial synchrony pattern in our paper, as they both indicate a decrease of correlation with distance. Nevertheless, there exists a fundamental difference between the two, i.e. the nature of correlation. The variogram has no temporal dimension, and the variance and correlation represent properties of the spatial distribution of the focal variable. However, in spatial synchrony, “correlation” always represents temporal correlation, e.g. correlation between two time series of productivity from two different locations. This temporal nature of correlation is essential to scale up our variability-based stability metric from local to regional scales. From spatial synchrony, our StAR approach goes one step further by clarifying the scaling patterns of stability or variability.

Regarding the originality of StAR, we have further checked the literature. We found a few earlier papers that hinted at the idea that stability might change with scale. Our StAR approach is thus consistent with these previous thoughts, but it provides a more quantitative framework to study the spatial scaling of stability. In the revised manuscript, we have added some sentences in the Introduction to explain this: *“The idea that stability changes with scale was first hinted at by Peterson et al.¹¹, who suggested that the resilience of ecosystems might increase with spatial scale due to scale-mediated effects of diversity. Another study used hierarchical theory to predict that stability at one hierarchical level, as measured by low variability, should increase proportionally to the number of lower-level components¹². The hierarchical levels used in this study can be interpreted either as discrete biological organizational levels (e.g. cell, organ, species) or as spatial scales (e.g. local ecosystem, landscape, region). Lastly, using a hierarchical partition of variability across spatial scales, Wang and Loreau¹³ predicted that temporal variability should generally decrease with area. But a quantitative framework has yet to be developed to study the spatial scaling of stability.”* Please see Page 3 Lines 43-53.

Finally, our study used spatio-temporal data of primary productivity and bird biomass to demonstrate that StARs apply to empirical data. These two datasets are, to our knowledge, among the best available for investigating large-scale spatio-temporal dynamics. They also represent two different taxa

(i.e. plants and birds) and two different data collection protocols (i.e. continuous and discrete grids). Moreover, they both agree with our models. To better clarify this, we have revised our abstract: “*When synchrony decays exponentially with distance, StARs exhibit three phases, characterized by steeper increases in stability at both small and large scales. Such triphasic StARs are observed for primary productivity from plot to continental scales. When synchrony decays as a power law with distance, StARs are quasilinear on a log-log scale. Such StARs are observed in North American birds*” (Page 2 Lines 26-30). We fully agree with the reviewer that more empirical studies are needed, as we have also discussed in our manuscript (Page 11 Lines 231-233). However, appropriate large-scale spatio-temporal data are still very limited. We hope that, with the recent development of new techniques (e.g. LiDAR), new data will soon become available for future spatio-temporal analyses. In the revised manuscript, we have added one sentence: “*Although large-scale spatio-temporal data are still very limited, the situation may improve with the development of new techniques such as airborne light detection and ranging (LiDAR)*” (Page 11 Lines 233-235).

I am also concerned about the loose definition and analysis in places. Firstly, the StAR should be defined more carefully via a formula. The current loose definition is not sufficient. Secondly, greater care should be taken in their explanation of multi-scale variability, where scale mismatches are a common source of error. Even the simplest spatio-temporal process has sampling-dependent parameters. These include the following:

1. Upper resolution scale. This is A.
2. Lower scale of resolution. Set to unity in this paper.
3. Inter-patch distance. Set to one in this paper (grid is flush in model)
4. Time-sampling behavior. In time too, ecological variables are multi-scale stochastic processes. Again there is a lower scale, an upper scale and a sampling rate. (Halley 2007).

Each of the above will affect how variability scales and hence the structure of the StAR. I found the explanation of these very scattered. The authors have clearly thought about the issues but they need to tell readers in a more structured way. The temporal issues in particular are skirted over very quickly. “temporal variance” is used (in Line 74) but not defined precisely. Stationarity is assumed in line-73, though the authors are aware that variance grows with observation time.

Response: Thank you for this insightful comment. We agree with the reviewer that variability depends on both spatial and temporal scales. While the effect of temporal scale has been relatively well studied in previous papers (Pimm & Redfearn 1988; Inchausti & Halley 2001; Halley 2007), the main focus of our study is on spatial scale. The reviewer clarifies three aspects of spatial scale by their points 1-3, i.e. extent, resolution, and sampling intensity. Since our StAR approach explores space continuously, the extent and resolution are represented by the largest and smallest area, respectively, on the x-axis of StAR. So, the reviewer’s concern on these two aspects is explicitly taken into account by the definition of StAR. For the third aspect, i.e. sampling intensity, our paper has examined StARs on both continuous landscapes (i.e. flush grids), in our model and in the primary productivity data, and non-continuous ones (i.e. spatially separated grids due to incomplete sampling), in the bird data. Moreover, with our theoretical model, we investigate the influence of sampling intensity and clarify that incomplete sampling could potentially increase the slope of StAR (please see Supplementary Note 2). In the revised manuscript, we have added some sentences to better clarify this: “*The sampling scheme can potentially influence the calculation of variability and hence the empirical patterns of StAR (see Supplementary Note 2). In the bird survey, although the hundreds of sampling routes cover the whole extent of eastern North America, the total*

sampling area represents only a relatively small proportion of the whole continent. Our theoretical model shows that incomplete spatial sampling could potentially increase the slope of StAR” (Page 8 Lines 169-173).

The reviewer points out that variability can similarly be affected by temporal scale, i.e. sampling resolution (i.e. “lower scale” in reviewer’s terminology), sampling intensity (i.e. “sampling rate”) and observation length (i.e. “upper scale”). With our bird data, we have investigated the effect of observation length and showed it can slightly alter the intercept and slope of StAR. But we had not explored the influence of sampling resolution and intensity. As for the sampling resolution, our study has fixed it to one year due to both data limitation (e.g. bird biomass data is collected once per year) and research interest (e.g. we are interested in the interannual dynamics of NPP, not seasonal oscillations). As for the sampling intensity, we fixed it to be annual. This is simply because we want to use most of available information in the data, given the relatively short time series in our data (i.e. NPP data: 15 year; bird data: 21 years). This said, we agree that such investigations may be useful for future studies on StAR. In the revised manuscript, we have added one sentence in the main text to acknowledge the potential influence of temporal sampling: *“The temporal sampling scheme may also influence the scaling patterns of stability (see Supplementary Note 2). With the bird data, we examined how observation length might affect the slope of StAR”* (Page 8 Lines 176-178). Furthermore, while concentrating our main text on spatial scale, we have extended our Supplementary Note 2 (“Sampling issues in StAR”) by adding two paragraphs to clarify potential sampling issues related to both spatial and temporal scales, as the reviewer highlights.

The reviewer also suggests to define a formula for StAR. For species-area relationship, people have used power ($S=c*A^z$) and logarithmic ($S=c+z*\log(A)$) functions to describe the dependence of biodiversity on area, which are well supported by data. For StAR, however, we still cannot conclude with some specific formula at this stage. The concept of StAR is being introduced in this paper and studied with two examples. Much more future research is required to clarify which function best describes StAR in empirical data. We will be most happy to see this come along in the future.

Finally, in the revised manuscript we have added two sentences to clarify the definition of “temporal variance” and “stationary”: *“All local patches are assumed to have identical temporal mean (μ) and variance (σ^2) of total biomass, i.e. the mean and variance of time series of total biomass. We assume the temporal dynamics are stationary and hence σ^2 is constant through time, although empirical data may exhibit non-stationary dynamics”*. Please refer to Page 12 Lines 261-264.

Finally, why is the word “stability” chosen, rather than variability? Stability is typically associated with dynamical systems, where it is an internal property of the ecological community. However, the StAR is a measure of variability that may simply be a linear response to externally imposed abiotic variability.

Response: The goal of a stability measure is to quantify the ability of a system to withstand perturbations. As such, stability reflects the interplay between intrinsic dynamical processes and the set of perturbations that act upon a system. Therefore, understanding the spatial scaling of stability implies understanding not only the scaling of intrinsic dynamical processes (e.g. species interactions, dispersal, etc...) but also that of external perturbations (e.g. climate events, fires, etc...). For this purpose, variability-based metrics, which measure an ecosystem's response to persistent and erratic environmental perturbations, offer an accessible starting point. Variability-based metrics have long been used as measures of stability, both theoretically (e.g. Ives et al. 1999; Lehman & Tilman 2000; Hughes & Roughgarden 2000; Ives & Carpenter 2007)

and empirically (e.g. Pimm & Redfearn 1988; Tilman 2006; Hector et al. 2010; Hautier et al. 2014). Indeed, a recent review found that variability was by far the most commonly used measure of stability in empirical studies (Donoghue et al. 2016). Variability-based metrics also have other merits: for instance, they can be defined consistently across levels of organization and scales (Tilman 2006; Hector et al. 2010; Wang & Loreau 2014). In the revised manuscript, we have added several sentences in the Introduction to better justify the use of variability-based stability metrics (please refer to Page 4 Lines 59-72).

On the other hand, we agree that variability represents only one of the many dimensions of stability, and different dimensions can be important in different contexts. Recent theoretical studies have demonstrated that variability is intrinsically related to other measures of stability such as asymptotic resilience (Ives 1995; Arnoldi et al. 2016), Holling's resilience (Scheffer et al. 2009), and persistence (Lande et al. 2003). As a consequence, our study based on temporal variability might also offer insights into the spatial scaling of other stability measures. Indeed, following the first reviewer's suggestion, we have developed a metapopulation model and demonstrated that our variability-based StAR had great potential as a tool to predict regime shifts in spatial ecological systems (see Page 11 Lines 236-245 in the main text and also Supplementary Note 4). This suggests that the spatial scaling of variability (i.e. our StAR) may be related to that of resilience. One promising direction for future research would be to compare and integrate StARs based on different stability metrics (e.g. variability, resilience, etc.), which may contribute to a multidimensional perspective on StARs and thereby provide a more comprehensive understanding of spatial scaling of stability. In the revised manuscript, we have added one sentence in the concluding paragraph that calls for new research on multidimensional StARs: "*As stability can be defined in multiple dimensions, a multidimensional StAR can be developed to achieve a more comprehensive understanding of spatial scaling of stability*" (Page 11 Lines 249-251).

Specific issues:

L-35. Ref Pimm (1991) or Pimm & Redfearn (1988). Important contribution to the dynamic nature of ecological variability.

Response: Thank you. We cited Pimm & Redfearn (1988).

L-91. The exponential model used is confusing due to the use of l(one) in one case and l(ell) in another. Which is it? Maybe an alternative lowercase letter for the grid address in Methods might be preferable.

Response: We are not sure about this confusion. We did not use "l(ell)" in this equation or any other equation in the manuscript.

L-192. "StARs could be used to predict the loss of population or ecosystem stability due to habitat destruction, in the same way as SARs". This claim also needs to be justified better.

Response: In the revised manuscript, we have added two sentences to better justify this point: "*For instance, a power function $S(A)=cA^z$ could be used to predict changes in bird population stability following habitat loss. Bird species exhibit large interspecific variations in their StARs, and disentangling their drivers would help us understand how different bird species may respond to habitat destruction differently.*" Please refer to Page 10 Lines 224-227.

Lines 214-224. This section is confusing. Most of the problems flow from the failure to define $S(A)$ precisely.

Response: In the revised manuscript, we have added one sentence to clarify the definition of $S(A)$: “We define ecosystem stability in a study area A ($S(A)$) as the reciprocal of squared coefficient of variation (CV^2) of biomass in area A : $S(A)=1/(CV^2(A))$ ” (Page 12 Lines 258-259).

References

1. Cressie, Noel. "Statistics for spatial data: Wiley series in probability and statistics." Wiley-Interscience, New York 15 (1993): 105-209.
2. Halley, John M. "How do Scale and Sampling Resolution Affect Perceived Ecological Variability and Redness?" in *The Impact of Environmental Variability on Ecological Systems*. Ed. D.A. Vasseur and K.S. McCann, Springer, Netherlands, 2007. 17-40.
3. Pimm, Stuart L. *The balance of nature?: ecological issues in the conservation of species and communities*. University of Chicago Press, 1991.
4. Pimm, Stuart L., and Andrew Redfearn. "The variability of population densities." *Nature* 334 (1988): 613-614.

Reviewer #4 (Remarks to the Author):

This manuscript proposes a measure of spatial stability for ecological systems. The manuscript then goes on to investigate various aspects of this measure both empirically and theoretically. I will separate my comments into two parts with the first focusing on what are essentially matters of presentation and the second on the underlying concepts.

I have real concerns with the presentation because I do not believe that the measure presented is a stability measure. Variability and stability are difference concepts. Stability should only relate (and is defined as) abilities to withstand perturbations and either remain close to some state (or return to the state). The variability of the system is determined by both the stability and the perturbations, and equating stability and variability (or more precisely the inverse of variability) is a very poor idea and is liable to lead to confusion. Thus, I would strongly argue against publication of the paper as is, leading to the question of whether something interesting and novel is contained in the underlying ideas of spatial variability as developed in the manuscript.

Response: The goal of a stability measure is to quantify the ability of a system to withstand perturbations. As such, stability reflects the interplay between intrinsic dynamical processes and the set of perturbations that act upon a system. Therefore, understanding the spatial scaling of stability implies understanding not only the scaling of intrinsic dynamical processes (e.g. species interactions, dispersal, etc...) but also that of external perturbations (e.g. climate events, fires, etc...). For this purpose, variability-based metrics, which measure an ecosystem's response to persistent and erratic environmental perturbations, offer an accessible starting point. Variability-based metrics have long been used as measures of stability, both theoretically (e.g. Ives et al. 1999; Lehman & Tilman 2000; Hughes & Roughgarden 2000; Ives & Carpenter 2007) and empirically (e.g. Pimm & Redfearn 1988; Tilman 2006; Hector et al. 2010; Hautier et al. 2014).

Indeed, a recent review found that variability was by far the most commonly used measure of stability in empirical studies (Donoghue et al. 2016). Variability-based metrics also have other merits: for instance, they can be defined consistently across levels of organization and scales (Tilman 2006; Hector et al. 2010; Wang & Loreau 2014). In the revised manuscript, we have added several sentences in the Introduction to better justify the use of variability-based stability metrics (please refer to Page 4 Lines 59-72).

On the other hand, we agree that variability represents only one of the many dimensions of stability, and different dimensions can be important in different contexts. Recent theoretical studies have demonstrated that variability is intrinsically related to other measures of stability such as asymptotic resilience (Ives 1995; Arnoldi et al. 2016), Holling's resilience (Scheffer et al. 2009), and persistence (Lande et al. 2003). As a consequence, our study based on temporal variability might also offer insights into the spatial scaling of other stability measures. Indeed, following the first reviewer's suggestion, we have developed a metapopulation model and demonstrated that our variability-based StAR had great potential as a tool to predict regime shifts in spatial ecological systems (see Page 11 Lines 236-245 in the main text and also Supplementary Note 4). This suggests that the spatial scaling of variability (i.e. our StAR) may be related to that of resilience. One promising direction for future research would be to compare and integrate StARs based on different stability metrics (e.g. variability, resilience, etc.), which may contribute to a multidimensional perspective on StARs and thereby provide a more comprehensive understanding of spatial scaling of stability. In the revised manuscript, we have added one sentence in the concluding paragraph that calls for new research on multidimensional StARs: "*As stability can be defined in multiple dimensions, a multidimensional StAR can be developed to achieve a more comprehensive understanding of spatial scaling of stability*" (Page 11 Lines 249-251).

Could an interesting paper be developed by instead focusing on an idea like spatial patterns of variability? I am also not convinced that this is possible. As far as I can tell, the theoretical development is simply a presentation of different patterns of spatial variability – the idea that variability relates to synchrony does not provide insights. Perhaps the most novel part of the paper is the empirical analysis of the spatial patterns of variability for different taxa. However, this likely can be simply explained by different dispersal distances.

Response: We do believe that the spatial scaling of variability can provide significant insights. Low variability is critical for the sustainable supply of ecosystem services (Pimm & Redfearn 1988; Schindler 2010). Thus, the spatial scaling of variability informs us about how the reliability of ecosystem services may change across scales. Moreover, the spatial scaling of variability (i.e. our StAR) might provide an indicator for regime shifts (i.e. Holling's resilience; see Scheffer, M. *et al.* 2009) in spatial ecological systems. During the revision of the manuscript, we have developed a metapopulation model and showed that, as the metapopulation approaches the regime shift, the intercept of StAR, its initial and asymptotic slopes, all decrease so that the triphasic shape gradually diminishes. Furthermore, we found that the decrease in the slope of StAR follows a specific pattern across scales. Spatial correlations propagate gradually through space, such that the initial slope of StAR decreases first, while the final slope decreases only close to the regime shift. Thus, StAR has great potential as a new tool to predict regime shifts in spatially structured systems. Please refer to the new paragraph in the revised discussion (Page 11 Lines 236-245) and the new Supplementary Note 4.

We respectfully disagree with the reviewer when they say that “the idea that variability relates to synchrony does not provide insights”. It is true that many previous studies have discussed the relation between the two patterns, but mostly in a qualitative way. To the best of our knowledge, our study is the first to demonstrate an explicit and quantitative link between them. For instance, we showed that light- and heavy-tail correlation functions could generate very different StARs (Figure 1). In particular, a light-tailed function (e.g. exponential decay of correlation with distance) can generate a triphasic StAR and the characteristic correlation length (or its square) determines its inflection point. By making the link between spatial patterns of synchrony and stability explicit, our study may provide a benchmark that people studying synchrony can use to understand the implications of their results for large-scale stability.

In our paper we investigated only two correlation functions, but they are general in the sense that they represent two classes of correlation patterns, i.e. light-tail and heavy-tail. We have explored several other correlation functions, and they result in qualitatively similar StARs. Their generality is also reflected by our data. The primary productivity and bird data represent very different taxa, scales, and landscape configurations. But their spatial synchrony and stability patterns are well described by our models.

Overall, we strongly believe that our work contributes to a quantitative understanding of synchrony and stability across scales. It calls for future quantitative research to explore spatial synchrony and disentangle the drivers of StARs in theory and data. In the Supplementary Note 3, we have summarized several potentially important drivers, including environmental correlation, dispersal, species distribution, and others. Exploring and understanding the way different mechanisms affect StARs provide new and exciting research prospects. Such new body of research will contribute to transforming the concept of StAR into a practical tool for understanding and predicting the long-term responses of species and ecosystems to habitat changes. In the revised manuscript, we have added several sentences in the concluding paragraph to clarify this. Please refer to Page 11 Lines 246-254.

Reference:

- Arnoldi, J. F., Loreau, M., & Haegeman, B. Resilience, reactivity and variability: A mathematical comparison of ecological stability measures. *J. Theor. Biol.* **389**, 47-59 (2016).
- Barros, C., Thuiller, W., Georges, D., Boulangeat, I. & Münkemüller, T. N-dimensional hypervolumes to study stability of complex ecosystems. *Ecol. Lett.* **19**, 729-742 (2016).
- Chisholm, R. A. & Pacala, S. W. Niche and neutral models predict asymptotically equivalent species abundance distributions in high-diversity ecological communities. *Proc. Natl. Acad. Sci. USA* **107**, 15821-15825 (2010).
- Cressie, N. *Statistics for spatial data*. (Wiley-Interscience, New York, 1993).
- Dakos, V., van Nes, E. H., Donangelo, R., Fort, H. & Scheffer, M. Spatial correlation as leading indicator of catastrophic shifts. *Theor. Ecol.* **3**, 163-174 (2010).
- Donohue, I. *et al.* On the dimensionality of ecological stability. *Ecol. Lett.* **16**, 421-429 (2013).
- Donohue, I. *et al.* Navigating the complexity of ecological stability. *Ecol. Lett.* **19**, 1172-1185 (2016).
- Halley, J. M. How do Scale and Sampling Resolution Affect Perceived Ecological Variability and Redness?. In *The Impact of Environmental Variability on Ecological Systems* (Springer Netherlands, 2007).

- Hanski, I., Zurita, G. A., Belloq, M. I. & Rybicki, J. Species–fragmented area relationship. *Proc. Natl. Acad. Sci. USA* **110**, 12715-12720 (2013).
- Hautier, Y. *et al.* Eutrophication weakens stabilizing effects of diversity in natural grasslands. *Nature* **508**, 521-525 (2014).
- Hector, A. *et al.* General stabilizing effects of plant diversity on grassland productivity through population asynchrony and overyielding. *Ecology* **91**, 2213-2220 (2010).
- Hubbell, S. P. *The unified neutral theory of biodiversity and biogeography*. (Princeton University Press, 2001).
- Hughes, J. B. & Roughgarden, J. Species diversity and biomass stability. *Am. Nat.* **155**, 618-627 (2000).
- Inchausti, P. & Halley, J. Investigating long-term ecological variability using the global population dynamics database. *Science* **293**, 655-657 (2001).
- Ives, A. R. (1995). Measuring resilience in stochastic systems. *Ecological Monographs* **65**, 217-233.
- Ives, A. R. & Carpenter, S. R. Stability and diversity of ecosystems. *Science* **317**, 58-62 (2007).
- Ives, A. R., Gross, K. & Klug, J. L. Stability and variability in competitive communities. *Science* **286**, 542-544. (1999).
- Jørgensen, S. E. & Nielsen, S. N. The properties of the ecological hierarchy and their application as ecological indicators. *Ecol. Indic.* **28**, 48-53 (2013).
- Lande, R., Engen, S. & Sæther, B. E. Spatial scale of population synchrony: environmental correlation versus dispersal and density regulation. *Am. Nat.* **154**, 271-281 (1999).
- Lande, R., Engen, S. & Saether, B. E. *Stochastic population dynamics in ecology and conservation* (Oxford Univ. Press, 2003).
- Lehman, C. L. & Tilman, D. Biodiversity, stability, and productivity in competitive communities. *Am. Nat.* **156**, 534-552 (2000).
- Levin, S. A. The problem of pattern and scale in ecology: the Robert H. MacArthur award lecture. *Ecology* **73**, 1943-1967 (1992).
- Liebhold, A., Koenig W.D. & Bjørnstad, O.N. Spatial synchrony in population dynamics. *Annu. Rev. Ecol. Evol. Syst.* **35**, 467-490 (2004).
- Lomolino, M. V. Ecology's most general, yet protean pattern: the species-area relationship. *J. Biogeogr* **27**, 17-26 (2000).
- Scheffer, M. *et al.* Early-warning signals for critical transitions. *Nature* **461**, 53-59 (2009).
- Peterson, G., Allen, C. R. & Holling, C. S. Ecological resilience, biodiversity, and scale. *Ecosystems* **1**, 6-18 (1998).
- Pimm, S. L. & Redfearn, A. The variability of population densities. *Nature* **334**, 613-614 (1988).
- Tilman, D., Reich, P. B. & Knops, J. M. Biodiversity and ecosystem stability in a decade-long grassland experiment. *Nature* **441**, 629-632 (2006).
- Wang, S. & Loreau, M. Ecosystem stability in space: α , β and γ variability. *Ecol. Lett.* **17**, 891-901 (2014).
- Wang, S. & Loreau, M. Biodiversity and ecosystem stability across scales in metacommunities. *Ecol. Lett.* **19**, 510-518 (2016).

Reviewers' Comments:

Reviewer #1 (Remarks to the Author):

Dear Authors,

I have read the revised ms and the rebuttal letter.

To my views, the authors have successfully address referees' main criticisms.

I am also happy to see that the authors have elaborated on some of my suggestions. In particular, in relation to what I said about the potential of StARs to characterize how far a system is from a regime shift, they have developed a model to precisely study, at least theoretically, the impact of critical transitions on the shape of StAR curves. I appreciate this addition very much.

In addition, they have revised and extended the ms in several other respects. No doubt authors' work will generate discussion in the field of ecology and interest also a wider audience.

I don't have further comments.

Reviewer #2 (Remarks to the Author):

The authors have responded satisfactory to my (and in my opinion the other referees) comments on the earlier draft, which have made the ms increase in readability and quality. I am therefore positive to publication in Nature Communications.

Reviewer #3 (Remarks to the Author):

This revised paper introduces and promotes a new quantity they call the Stability Area Relation. This version contains some revisions (various additions of paragraphs, rewording etc) without much substantive change in content. In my opinion this m.s. remains a novel ideas paper with promise to be shown correct by subsequent results. And it should be called the variability area relation.

In their replies, the authors are pretty up-front about the scope and limitations of their paper (to Referee-1 "We fully agree with the value of doing this, but we believe that a new significant

project is required” and to me “Much more future research is required to clarify which function best describes StAR in empirical data. We will be most happy to see this come along in the future”). For this reason, though, I would say it is a sound and useful paper but the relation does not yet have enough material or depth to warrant publication in NComms.

Reading also the concerns of Referee-4, I have become more convinced that the choice of “stability” in the title (and throughout) is inappropriate. The quantity measured in this paper is variability, since in Eq. (1) $StAR=1/CV^2$. But variability is not necessarily the same as stability. Yes, variability is widely used as a proxy for stability which is appropriate in specific contexts such as Lotka-Volterra models and their extensions. Pimm & Redfearn, who also measured variability, wrote that “population variability is one of several meanings of ecological stability”. And that’s why their paper is entitled the “The variability of population densities” rather than “The stability of population densities”. Would it be wrong to use “variability”? Or would a VarAR be somehow different from a StAR?

Reviewer #4 (Remarks to the Author):

I stand by my earlier review and would reiterate that variability is a consequence of an interaction between perturbations and dynamical properties, but is definitely not stability. I do not see any improvement in the manuscript.

Reviewers' comments:

Reviewer #1 (Remarks to the Author):

Dear Authors,

I have read the revised ms and the rebuttal letter.

To my views, the authors have successfully address referees' main criticisms.

I am also happy to see that the authors have elaborated on some of my suggestions. In particular, in relation to what I said about the potential of StARs to characterize how far a system is from a regime shift, they have developed a model to precisely study, at least theoretically, the impact of critical transitions on the shape of StAR curves. I appreciate this addition very much.

In addition, they have revised and extended the ms in several other respects. No doubt authors' work will generate discussion in the field of ecology and interest also a wider audience.

I don't have further comments.

Response: Thank you for your supportive comments.

Reviewer #2 (Remarks to the Author):

The authors have responded satisfactory to my (and in my opinion the other referees) comments on the earlier draft, which have made the ms increase in readability and quality. I am therefore positive to publication in Nature Communications.

Response: Thank you.

Reviewer #3 (Remarks to the Author):

This revised paper introduces and promotes a new quantity they call the Stability Area Relation. This version contains some revisions (various additions of paragraphs, rewording etc) without much substantive change in content. In my opinion this m.s. remains a novel ideas paper with promise to be shown correct by subsequent results. And it should be called the variability area relation.

Response: We have changed our terminology from "stability" to "invariability" to make our manuscript technically more precise. This includes re-naming our

theory “Invariability-Area Relationship (IAR)”, as you have suggested.

In their replies, the authors are pretty up-front about the scope and limitations of their paper (to Referee-1 “We fully agree with the value of doing this, but we believe that a new significant project is required” and to me “Much more future research is required to clarify which function best describes StAR in empirical data. We will be most happy to see this come along in the future”). For this reason, though, I would say it is a sound and useful paper but the relation does not yet have enough material or depth to warrant publication in NComms.

Response: We strongly believe that our paper provides a novel and significant contribution: we define and describe the Invariability-Area Relationship (IAR) for the first time, provide a theoretical foundation for the IAR, quantify it using two data sets, and explore its implications for understanding the impacts of habitat loss and predicting regime shifts. We acknowledge we have not solved all the possible issues that may arise, but this only points to the wide variety of questions of interest for which this relationship is relevant and will be useful. In the opinion of referee 1, this paper will “generate discussion in the field of ecology and interest also a wider audience”.

We clearly acknowledge the scope of our study in the manuscript. For instance, in the Discussion we have clearly stated that future research are needed both theoretically, e.g. to explore the scaling of other stability metrics and clarify their drivers, and empirically, e.g. to investigate IAR for different taxa and landscape configurations.

Reading also the concerns of Referee-4, I have become more convinced that the choice of “stability” in the title (and throughout) is inappropriate. The quantity measured in this paper is variability, since in Eq. (1) $StAR=1/CV^2$. But variability is not necessarily the same as stability. Yes, variability is widely used as a proxy for stability which is appropriate in specific contexts such as Lotka-Volterra models and their extensions. Pimm & Redfearn, who also measured variability, wrote that “population variability is one of several meanings of ecological stability”. And that’s why their paper is entitled the “The variability of population densities” rather than “The stability of population densities”. Would it be wrong to use “variability”? Or would a VarAR be somehow different from a StAR?

Response: We fully agree that “variability is one of several meanings of ecological stability”. Indeed, we have clearly acknowledged this in our Introduction and Discussion. To make our manuscript technically more precise, we have changed our terminology to “invariability” in the revised manuscript. More specifically, we changed the term “Stability-Area Relationship (StAR)” into “Invariability-Area Relationship (IAR)” throughout the paper and changed “stability” into “invariability” in our Methods and Results sections, as well as in

all specific statements in the Introduction and Discussion sections.

However, we still feel important to discuss the IAR in the general context of ecological stability, because scientists and managers are not interested in variability per se, but in what it tells about the stability of ecological systems. In the revised manuscript, we have restructured our Introduction to better clarify why invariability offers an appropriate starting point for studying the spatial scaling of stability (please refer to Page 4 Line 64-78). We have also added some sentences in the Discussion to clarify that future research is needed to explore the spatial scaling of other stability measures (in particular, asymptotic resilience) in order to achieve a more comprehensive understanding of spatial scaling of stability (please refer to Page 12 Line 251-255).

Reviewer #4 (Remarks to the Author):

I stand by my earlier review and would reiterate that variability is a consequence of an interaction between perturbations and dynamical properties, but is definitely not stability. I do not see any improvement in the manuscript.

Response: We would like to reiterate that the inverse of temporal variability, which we call invariability, is a standard and widely accepted measure of temporal stability in ecology. Classic reviews (Pimm 1984; McCann 2000; Ives & Carpenter 2007) have always listed variability or invariability as one of the measures of stability. The Millennium Ecosystem Assessment (2005), which represents the view of a large number of experts worldwide, used invariability when talking about stability. According to a recent comprehensive review, variability-based metrics have been used to quantify stability in 61% of all experimental and 72% of all observational papers (~180 in total) that have been published in “three high-impact multidisciplinary journals and four leading general ecology journals: Nature, Science, PNAS, Ecology Letters, Ecology, Oikos and American Naturalist” (Donohue et al. 2016). These studies included many, if not most, of the world’s largest biodiversity and global change experiments (e.g. Tilman 2006; Worm et al. 2006; Hector et al. 2010; Cardinale et al. 2012; Hautier et al. 2014). This trend is continuing and even increasing, as evidenced by several recent papers that have used variability as the sole measure of stability and that have been published in high-impact journals (e.g. Hautier et al. 2014 Nature; Hautier et al. 2015 Science; Prieto et al. 2015 Nature Plants; Bluthgen et al. 2016 Nature Communications; Shi et al. 2016 Nature Communications). So we believe that the view that variability is not a stability measure does not represent the current consensus in the discipline.

We understand that this view may arise from the fact that theoretical and empirical ecologists have interests in different stability metrics. In contrast to the empirical focus on variability, the review also showed that 57% of theoretical papers (~180 in total) focused on asymptotic stability (Donohue et al. 2016). This separation has challenged ecologists to reconcile theory and data and calls

for a synthesis. Although asymptotic stability is easy to investigate mathematically, it is difficult to quantify empirically (the review showed only 4% of empirical papers studied this metric). The alternative is to develop a theory of invariability. The latter approach has been much more fruitful and has offered novel insights into empirical data (the review shows that 18% of theoretical papers, including some influential recent papers, studied variability; e.g. Hughes & Roughgarden 1998; Ives et al. 1999; Lehman & Tilman 2000; Loreau and de Mazancourt 2013). Moreover, theoreticians have also tried to clarify the nature of different stability metrics and to unify them mathematically. For instance, the work of some of my co-authors showed that invariability could be defined flexibly across levels of organization (e.g. ecosystem, average species, rare species) and that the invariability of the rarest species is intrinsically related to asymptotic stability (Arnoldi et al. 2016; Haegeman et al. 2016) and structural stability (Arnoldi and Haegeman 2016). In other words, invariability offers an opportunity to unify different stability metrics.

Our study is a combination of theory and data. To this end, invariability offers an appropriate starting point. This said, we totally agree with reviewer 3 and Pimm & Redfearn (1988) that “variability is one of several meanings of ecological stability”. Indeed, we have clearly acknowledged this in our Introduction and Discussion. We have also explored the link between invariability and other stability measures (e.g. regime shift) and called for future studies to integrate different stability metrics in a spatial context.

In conclusion, we feel that using invariability as our metric of stability is a perfectly reasonable and acceptable choice that agrees with much of the ecological literature. This said, since we agree that invariability is but one of the meanings of ecological stability, we have revised our manuscript to make it technically more precise by changing the term “Stability-Area Relationship (StAR)” into “Invariability-Area Relationship (IAR)” throughout the paper and by changing “stability” into “invariability” in our Methods and Results sections (as well as in all specific statements in the Introduction and Discussion sections). But we still feel important to discuss the IAR in the general context of ecological stability, because scientists and managers are not interested in variability per se, but in what it tells about the stability of ecological systems.

In the revised manuscript, we have also restructured the Introduction to better clarify why invariability-based metrics offer an appropriate starting point for studying spatial scaling of stability (Page 4 Line 64-78). Lastly, we have also added some sentences in the Discussion to clarify that theoretical research has mainly focused on asymptotic resilience and extending our IAR approach to the asymptotic resilience, as well as other stability metrics, will contribute to a more comprehensive understanding of spatial scaling of stability (Page 12 Line 251-255).

References:

Arnoldi, J. F., & Haegeman, B. Unifying dynamical and structural stability of

- equilibria. *Proc. R. Soc. A* 472, 20150874 (2016)
- Arnoldi, J. F., Loreau, M., & Haegeman, B. Resilience, reactivity and variability: A mathematical comparison of ecological stability measures. *J. Theor. Biol.* 389, 47-59 (2016).
- Blüthgen, N. et al. Land use imperils plant and animal community stability through changes in asynchrony rather than diversity. *Nature Communications* 7, 10697 (2016).
- Cardinale, B. et al. Biodiversity loss and its impact on humanity. *Nature* 486, 59-67. (2012)
- Donohue, I. et al. Navigating the complexity of ecological stability. *Ecol. Lett.* 19, 1172-1185 (2016).
- Haegeman, B., Arnoldi, J. F., Wang, S., de Mazancourt, C., Montoya, J. M., & Loreau, M. Resilience, invariability, and ecological stability across levels of organization. *bioRxiv*, 085852 (2016).
- Hautier, Y. et al. Eutrophication weakens stabilizing effects of diversity in natural grasslands. *Nature* 508, 521-525 (2014)
- Hautier, Y., Tilman, D., Isbell, F., Seabloom, E. W., Borer, E. T., & Reich, P. B. Anthropogenic environmental changes affect ecosystem stability via biodiversity. *Science* 348, 336-340. (2015).
- Hector, A. et al. General stabilizing effects of plant diversity on grassland productivity through population asynchrony and overyielding. *Ecology* 91, 2213-2220 (2010).
- Hughes, J. B., & Roughgarden, J. (1998). Aggregate community properties and the strength of species' interactions. *Proceedings of the National Academy of Sciences*, 95(12), 6837-6842.
- Ives, A. R. & Carpenter, S. R. Stability and diversity of ecosystems. *Science* 317, 58-62 (2007).
- Ives, A. R., Gross, K. & Klug, J. L. Stability and variability in competitive communities. *Science* 286, 542-544. (1999).
- Lehman, C. L. & Tilman, D. Biodiversity, stability, and productivity in competitive communities. *Am. Nat.* 156, 534-552 (2000).
- Loreau, M., & Mazancourt, C. Biodiversity and ecosystem stability: a synthesis of underlying mechanisms. *Ecol. Lett.* 16, 106-115. (2013).
- McCann, K. S. The diversity-stability debate. *Nature* 405, 228-233 (2000).
- Millennium Ecosystem Assessment. *Ecosystems and Human Well-Being: Biodiversity Synthesis*. Washington, DC: Island Press (2005).
- Pimm, S. L. The complexity and stability of ecosystems. *Nature* 307, 321-326 (1984).
- Pimm, S. L. & Redfearn, A. The variability of population densities. *Nature* 334, 613-614 (1988).
- Prieto, I. et al. (2015). Complementary effects of species and genetic diversity on productivity and stability of sown grasslands. *Nature Plants* 1, 15033.
- Shi, Z. et al. Dual mechanisms regulate ecosystem stability under decade-long warming and hay harvest. *Nature Communications* 7, 11973 (2016).

Tilman, D., Reich, P. B. & Knops, J. M. Biodiversity and ecosystem stability in a decade-long grassland experiment. *Nature* 441, 629-632 (2006).

Worm, B. et al. Impacts of biodiversity loss on ocean ecosystem services. *Science* 314, 787-790 (2006)